# Optimization of Eugenol, Camphor, and Terpineol Mixture Using Simplex-Centroid Design for Targeted Inhibition of Key Antidiabetic Enzymes

**DOI:** 10.3390/cimb47070512

**Published:** 2025-07-02

**Authors:** Amine Elbouzidi, Mohamed Jeddi, Abdellah Baraich, Mohamed Taibi, Mounir Haddou, Naoufal El Hachlafi, Meryem Idrissi Yahyaoui, Reda Bellaouchi, Bouchra El Guerrouj, Khalid Chaabane, Mohamed Addi

**Affiliations:** 1Laboratoire d’Amélioration des Productions Agricoles, Biotechnologie et Environnement (LAPABE), Faculté des Sciences, Université Mohammed Premier, Oujda 60000, Morocco; mohamedtaibi9@hotmail.fr (M.T.); mounir.haddou.d23@ump.ac.ma (M.H.); elguerroujb@gmail.com (B.E.G.); k.chaabane@ump.ac.ma (K.C.); 2Laboratory of Microbial Biotechnology and Bioactive Molecules, Faculty of Sciences and Technologies, Sidi Mohamed Ben Abdellah University, Imouzzer Road, Fez P.O. Box 2202, Morocco; mohamed.jeddi@usmba.ac.ma; 3Laboratory of Bioresources, Biotechnology, Ethnopharmacology and Health, Faculty of Sciences, Mohammed First University, Boulevard Mohamed VI, B.P. 717, Oujda 60000, Morocco; abdellah.baraich@ump.ac.ma (A.B.); iy.meryem@ump.ac.ma (M.I.Y.); 4Centre de l’Oriental des Sciences et Technologies de l’Eau et de l’Environnement (COSTEE), Université Mohammed Premier, Oujda 60000, Morocco; r.bellaouchi@ump.ac.ma; 5Faculty of Medicine and Pharmacy, Ibn Zohr University, Guelmim 81000, Morocco; naoufal.elhachlafi@usmba.ac.ma

**Keywords:** mixture design, eugenol, camphor, terpineol, bioavailability, IC_50_ optimization, combined effect, natural compounds

## Abstract

The optimization of bioactive compound mixtures is critical for enhancing pharmacological efficacy. This study investigates, for the first time, the combined effects of eugenol, camphor, and terpineol, focusing on their half-maximal inhibitory concentrations (IC_50_) across multiple biological responses related to diabetes management. Using a mixture design approach, the objective was to determine the optimal formulation that maximizes bioactivity and validate the findings experimentally. A simplex-centroid design was applied to evaluate the combined effects of eugenol, camphor, and terpineol on AAI _IC50_, AGI _IC50_, LIP _IC50_, and ALR _IC50_ responses. The desirability function was used to determine the ideal composition. The optimized formulation was experimentally validated using in vitro assays, and IC50 values were measured for each response using standard protocols. Results: The optimal formulation identified was 44% eugenol, 0.19% camphor, and 37% terpineol, yielding IC_50_ values of 10.38 µg/mL (AAI), 62.22 µg/mL (AGI), 3.42 µg/mL (LIP), and 49.58 µg/mL (ALR). The desirability score (0.99) confirmed the effectiveness of the optimized blend. Experimental validation of the optimal mixture resulted in IC_50_ values of 11.02 µg/mL (AAI), 60.85 µg/mL (AGI), 3.75 µg/mL (LIP), and 50.12 µg/mL (ALR), showing less than 10% deviation from predicted values, indicating high model accuracy. This study confirms the combined potential of eugenol, camphor, and terpineol, with eugenol and terpineol significantly enhancing bioactivity. The validated formulation demonstrates potential for pharmaceutical and cosmeceutical applications. Future research should explore mechanistic interactions, bioavailability, and in vivo efficacy to support the development of optimized natural compound-based therapies.

## 1. Introduction

*Diabetes mellitus* (DM) is a chronic metabolic disorder characterized by persistent hyperglycemia resulting from defects in insulin secretion, insulin action, or both [1]. The management of diabetes often involves the inhibition of key enzymes such as α-amylase, α-glucosidase, lipase, and aldose reductase, which play significant roles in carbohydrate and lipid metabolism [2,3]. Inhibiting these enzymes can help regulate postprandial blood glucose levels and mitigate complications associated with diabetes [1]. DM has seen a significant global increase in prevalence over recent decades. As of 2024, the World Health Organization (WHO) reports that over 800 million adults worldwide are living with diabetes, a figure that has more than quadrupled since 1990. Projections suggest that the global burden of diabetes will continue to rise. The International Diabetes Federation (IDF) estimates that by 2045, approximately 783 million adults will be living with diabetes, representing a 46% increase from current numbers [4].

Natural compounds have garnered considerable attention for their potential therapeutic effects in diabetes management due to their diverse pharmacological properties and lower side effects compared to synthetic drugs [5]. Eugenol, camphor, and terpineol are bioactive monoterpenes found in various essential oils, and their antidiabetic properties have been the focus of numerous studies. Eugenol, a key component of clove oil, has been reported to effectively lower blood glucose levels by inhibiting α-glucosidase activity, an enzyme involved in carbohydrate digestion, and by preventing the formation of advanced glycation end-products (AGEs), which are implicated in diabetes complications [6]. Eugenol has been shown to decrease glucose levels, improve lipid profiles, and reduce oxidative stress in hyperglycemic rodents. It also ameliorates renal and hepatic damage and restores antioxidant defense systems, although it does not affect insulin levels directly [6]. In diabetic rats, eugenol reduces serum glucose, triglycerides, and cholesterol, and enhances insulin sensitivity by activating the GLUT4-AMPK signaling pathway [7].

Similarly, camphor, a compound found in herbs like rosemary (*Rosmarinus officinalis* L.) and the camphor tree (*Camphora officinarum* Nees.), has shown promise in modulating glucose metabolism. Camphor has demonstrated hypoglycemic activity by lowering blood glucose, cholesterol, and triglyceride levels in diabetic rats. It also increases HDL-cholesterol levels [8]. The antidiabetic effects of camphor are largely due to its antioxidant properties, which enhance the activity of enzymes like glutathione peroxidase, catalase, and superoxide dismutase, thereby reducing oxidative stress in liver, pancreas, and kidney tissues [8]. Terpineol, a monoterpene alcohol found in essential oils like those from *Afromomum* species, has shown potential in inhibiting enzymes like α-amylase and α-glucosidase, which are involved in carbohydrate metabolism, suggesting its role in managing type-2 diabetes (T2DM) [9].

While individual studies have underscored the antidiabetic potential of eugenol, camphor, and terpineol, limited research has explored their combined effects. Synergistic interactions among these compounds could lead to enhanced therapeutic efficacy, which remains an underexplored area of research. Statistical methods like the simplex-centroid design provide a powerful approach to optimize mixtures by evaluating the combined effects of different components [10,11]. This design facilitates the systematic investigation of various combinations, enabling the identification of the optimal mixture for desired therapeutic outcomes.

This study aims to employ a simplex-centroid design to optimize the mixture of eugenol, camphor, and terpineol for the inhibition of key antidiabetic enzymes. By systematically evaluating different combinations of these compounds, this research seeks to identify a formulation that maximally inhibits α-amylase, α-glucosidase, lipase, and aldose reductase activities. These enzymes are critical in glucose and lipid metabolism, and their inhibition is a promising strategy for diabetes management. The findings from this study could contribute to the development of novel, natural therapeutic agents that offer a safer and more effective alternative for managing diabetes.

## 2. Materials and Methods

### 2.1. Molecules and Enzymes

Eugenol, camphor, and α-terpineol were all of analytical grade, and were purchased from Sigma Aldrich, Darmastadt, Germany. α-glucosidase enzyme from *Saccharomyces cerevisiae*, α-amylase from porcine pancreas, and porcine pancreatic lipase were purchased from Sigma Aldrich. Aldose Reductase Inhibitor Screening Kit (Biovision, Milpitas, CA, USA) was used for the aldose reductase inhibition assay.

### 2.2. Antioxidant Activity

#### 2.2.1. ABTS

The ABTS assay was determined according to Elbouzidi et al. [12], with some modifications. The stock solutions included 7.4 mM ABTS and 2.6 mM potassium persulfate. ABTS radical cation was prepared by mixing the two stock solutions in equal quantities and allowed to stand for 16 h at room temperature in the dark. And then, ABTS radical solution was adjusted with phosphate-buffered saline (pH 7.4) to an absorbance of 0.7–0.8 at 734 nm. For this assay, 190 μL of ABTS radical solution was mixed with 10 μL of each test sample solution in 96-well plates. After 60 min, the decrease of absorbance was measured at 734 nm using a microplate reader (Thermo Fisher Scientific Inc., Waltham, MA, USA). The standard curve was linear between 0.025 and 0.9 mM Trolox. Results were expressed in mM Trolox equivalent per mL of solution. Additional dilution was needed if the ABTS value measured was over the linear range of the standard curve.

#### 2.2.2. Ferric-Reducing Antioxidant Power (FRAP)

The antioxidant capacity of the tested compounds was evaluated using the ferric-reducing antioxidant power (FRAP) assay, based on the reduction of ferric-tripyridyltriazine (Fe^3+^-TPTZ) complex to its ferrous form (Fe^2+^), resulting in a blue-colored product with a maximum absorbance at 593 nm. The assay was performed according to the modified method of Benzie and Strain [13], with modifications for microplate format and adjusted concentration ranges according to Rodrigues et al. [14]. The FRAP working solution was freshly prepared by mixing 300 mM acetate buffer (pH 3.6), 10 mM TPTZ in 40 mM HCl, and 20 mM FeCl_3_·6H_2_O in a 10:1:1 volume ratio. The mixture was warmed to 37 °C before use to ensure optimal reaction conditions. For each reaction, 180 µL of the freshly prepared FRAP reagent was added to a well of a 96-well microplate, followed by 20 µL of the test compound at different concentrations (1–200 µg/mL) prepared in methanol. The mixture was incubated at 37 °C for 30 min. The increase in absorbance was measured at 593 nm using a microplate reader. Trolox was used as a reference antioxidant, and the results were expressed as EC_50_ values, defined as the concentration of compound required to achieve 50% of the maximal ferric-reducing capacity. All measurements were performed in triplicate and presented as the mean ± standard deviation.

#### 2.2.3. Cupric Ion-Reducing Antioxidant Capacity Assay (CUPRAC)

The total antioxidant capacity of the tested monoterpenes was also assessed using the Cupric ion-reducing antioxidant capacity (CUPRAC) assay, which measures the ability of antioxidants to reduce the Cu^2+^-neocuproine complex to the stable-colored Cu^+^-neocuproine chelate, absorbing at 450 nm. The assay was carried out following the protocol provided by the CUPRAC assay kit (Bioquochem, Asturias, Spain), with slight modifications to accommodate the tested compounds according to Salem et al. [15]. The CUPRAC working solution consisted of a mixture of copper (II) chloride (10 mM), neocuproine (7.5 mM), and ammonium acetate buffer (1 M, pH 7.0) in a 1:1:1 volume ratio. For the assay, 40 µL of each test solution (prepared in methanol at concentrations ranging from 1 to 200 µg/mL) was added to 200 µL of the CUPRAC working solution in a well of a 96-well plate. The plate was incubated at room temperature for 30 min in the dark to allow full color development. The absorbance was then measured at 450 nm using a microplate spectrophotometer (Thermo Fisher Scientific Inc., Waltham, MA, USA). Trolox was used as a positive control, and antioxidant capacity was expressed as IC_50_ values, representing the concentration at which 50% of maximal cupric-reducing activity was achieved. All tests were conducted in triplicate, and results are presented as mean ± SD.

### 2.3. Diabetes Enzymes Inhibitory Activities

#### 2.3.1. α-Amylase Inhibition Assay

Inhibition of the compounds, eugenol, camphor, and terpineol, on the α-amylase enzyme activity was determined kinetically using acarbose as a standard [16,17]. A pre-incubation mixture was prepared which contained 40 mM phosphate buffer (pH 6.9) together with different concentrations (1, 5, 10, 50, 100, 200 µg/mL) of the compounds. A 0.2 U of α-amylase enzyme was added to this mixture, and the plate was then incubated for 15 min at 37 °C. Then, 2-chloro-4-nitrophenyl maltotrioside (CNPG3) substrate was added, where the final concentration was 0.9 µmole/mL. The increase in absorbance at 405 nm, which is proportional to the enzyme activity, was recorded for 3 min. A control which was devoid of the compounds was also run in parallel. The 50% inhibitory concentration (IC_50_) was determined and presented for each compound, mixture, and the standard (acarbose).

#### 2.3.2. α-Glucosidase Inhibition Assay

The activity of the compounds was assessed spectrophotometrically according to Li et al. [18], with slight modifications. In a volume of 100 µL of 40 mM phosphate buffer (pH 6.9), different concentrations (1, 5, 10, 50, 100, 200 µg/mL) of the compounds were prepared. The tested samples were added to a reaction mixture in a 96-well microplate containing 20 µL of the phosphate buffer and 20 µL of 5 mM *p*-nitrophenyl α -D-glucopyranoside (PNP-G) in the buffer. Then 20 µL of 0.4 U/mL α-glucosidase in the buffer was added, and the plate was mixed and incubated at 37 °C for 15 min. After incubation, the absorbance was measured at 405 nm in the microplate reader. Acarbose was used as a reference compound, and a control was performed without the test samples. The IC_50_ for test compounds as well as the standard was calculated and presented.

#### 2.3.3. Lipase Inhibition Assay

The porcine pancreatic lipase inhibition assay was utilized in this work, with modifications made based on a previous study [19,20]. A stock solution of each compound at 1 mg/mL in 10% DMSO was used to prepare five distinct solutions with concentrations 1, 5, 10, 50, 100, 200 µg/mL by means of dilution with methanol. Just before usage, the pancreatic lipase enzyme was freshly produced as a 1 mg/mL stock solution. Additionally, a stock solution of PNPB (*p*-nitrophenyl butyrate) was created by dissolving 20.9 mg of PNPB in 2 mL of acetonitrile. In individual test tubes, 0.1 mL of porcine pancreatic lipase (1 mg/mL) was mixed with 0.2 mL of each compound concentration. To achieve a final volume of 1 mL, the Tri-HCl solution (pH 7.4) was added, and the mixtures were incubated at 25 °C for 15 min. Following the incubation, 0.1 mL of PNPB solution was introduced to each test tube. The mixtures were once again incubated for 30 min at 37 °C. The assessment of pancreatic lipase activity involved measuring the hydrolysis of *p*-nitrophenyl butyrate to *p*-nitrophenol at 405 nm using a UV-visible spectrophotometer (JASCO Corporation, Hachioji-shi, Tokyo, Japan). The same procedure was replicated for orlistat, which served as a positive control, with identical concentrations as mentioned earlier. All tests were conducted in triplicate, and the inhibition was calculated as per Equation (1).(1)$$\%Lipase\;Inhibition=\left[\frac{\left(AB-AE\right)}{AB}\right]\times 100$$
where AB is the recorded absorbance of the blank solution, and AE is the recorded absorbance of the compound’s solution.

#### 2.3.4. Aldose Reductase Activity

Determination of the aldose reductase inhibitory effectiveness of the investigated compounds was performed kinetically using the Aldose Reductase Inhibitor Screening Kit (Biovision, Milpitas, CA, USA), according to the manufacturer’s protocol. The decrease in the absorption of NADPH at 340 nm was measured in a mixture containing NADPH, the enzyme, and its substrate, DL-glyceraldeyde, in addition to test samples [21]. Different concentrations of the compounds (1, 5, 10, 50, 100, 200 µg/mL) were used, and quercetin was the reference standard in the assay. Briefly, 60 μL of NADPH was added to all wells containing 10 μL of either the enzyme assay buffer, test samples, or quercetin. A total of 90 μL of freshly prepared Aldose Reductase Solution was then added, and the plate was incubated protected from light at 37 °C for 15–20 min. After incubation, 40 μL of enzyme substrate in assay buffer was added, and the absorbance was measured at 340 nm kinetically for 60–90 min at 37 °C. A background control free from the enzyme was run and subtracted from the readings. The IC_50_ was calculated and presented for the samples and the standard.

### 2.4. Experimental Design

#### 2.4.1. Mixture Design

An augmented simplex-centroid design, as outlined by Benkhaira et al. [22] and Elbouzidi et al. [11], was employed to optimize the antidiabetic potential of the combined molecules of eugenol, camphor, and terpineol. This experimental methodology allows for a systematic evaluation of mixture proportions to determine the most effective combination. The composition details of the molecule system are presented in Table 1, where each molecule’s proportion was varied between 0 and 1, ensuring the total proportions of the three molecules always summed to 1. The antidiabetic activity of these mixtures was evaluated against four critical enzymes involved in diabetes management: α-amylase, α-glucosidase, lipase, and aldose reductase. This assessment provided valuable insights into the combined or individual effects of these molecules in inhibiting these enzymes. This approach not only facilitates precise formulation but also underscores the potential of molecule combinations in effectively addressing diabetes-related challenges.

#### 2.4.2. Experimental Matrix and Mathematical Model

In this study, 10 experimental trials were designed and mapped onto an equilateral triangle (Figure 1), which visually represents the varying proportions of the components being investigated. The three pure components are situated at the triangle’s vertices (M_1_, M_2_, M_3_). Binary mixtures, where two components are combined in equal proportions (0.5/0.5), are positioned at the midpoints of the triangle’s edges (M_4_, M_5_, M_6_). A ternary mixture with equal proportions of all three molecules (0.33/0.33/0.33) is located at the centroid of the triangle (M_7_). To ensure consistency and validate the results, the experiment was conducted three times, including three control points (M_10_, M_11_, M_12_) representing ternary mixtures with varying proportions (0.66/0.17/0.17).

A cubic model was applied to describe the responses, incorporating the independent variables. This approach facilitated a detailed analysis of the interactions between components, with the responses represented by the following mathematical Equation (2):**Y = δ_1_M_1_ + δ_2_M_2_ + δ_3_M_3_ + δ_12_M_1_M_2_ + δ_13_M_1_M_3_ + δ_23_M_2_M_3_ + δ_123_M_1_M_2_M_3_ + ε**(2)

In this equation, Y denotes the experimental response, measured as the IC_50_ (in µg/mL). The coefficients δ_1_, δ_2_, and δ_3_ represent the linear effects of each individual component. The binary interaction effects between pairs of components are indicated by the coefficients δ_12_, δ_13_, and δ_23_, while δ_123_ reflects the interaction effect of the ternary combination. The term ε (epsilon) accounts for regression error, capturing any variation not explained by the model. This formulation enables the quantification of both individual and interactive contributions of the components to the overall response.

### 2.5. Statistical Analysis

The adequacy of the model was evaluated by comparing the mean square lack of fit (MSLOF) to the mean square pure error (MSPE). Higher MSLOF/MSPE values may indicate potential deficiencies in the model’s fit. The coefficient of determination (R^2^) was also used to assess the model’s quality, reflecting its ability to explain variability in the data. The statistical validity of the mathematical model was tested at a 95% confidence level using the F-ratio, calculated as the ratio of the mean square regression (MSR) to the mean square residual (MSr). A higher F-value indicates that the model explains a greater portion of the variability. The significance of individual factors was determined using the Student’s *t*-test, and the overall model significance was confirmed through an F-test in an analysis of variance (ANOVA). All statistical analyses were performed using Design-Expert software (version 12) and SAS JMP^®^ (version 14), with results expressed as means ± standard deviation (SD) from three independent replicates (*n* = 3).

### 2.6. Optimization Tools

To optimize the system, contour and 3D surface plots were utilized to visualize trade-off regions among the components, aiding in the identification of optimal mixtures. A desirability function was applied to determine the most favorable outcomes, balancing the studied factors. This function scales results within a range from 0 to 1, where 0 represents an undesirable outcome and 1 indicates a highly desirable outcome. This approach ensures a practical and efficient optimization process, enabling the identification of formulations that maximize system performance.

## 3. Results and Discussion

### 3.1. Antioxidant Activity of Eugenol, Camphor, and Terpineol

The antioxidant activity of eugenol, camphor, and terpineol was systematically assessed using three complementary assays: ABTS, FRAP, and CUPRAC, as illustrated in Figure 2.

The ABTS assay (Figure 2A) quantifies the ability of tested compounds to neutralize ABTS•^+^ radicals, providing an indicator of their free radical-scavenging potential. Among the investigated compounds, eugenol exhibited the highest antioxidant activity, as demonstrated by its lowest IC_50_ value (19.42 ± 0.58 µg/mL). In contrast, camphor displayed the weakest radical-scavenging ability, with an IC_50_ of 755.82 ± 67.30 μg/mL, whereas terpineol showed moderate activity (IC_50_ = 476.49 ± 19.93 μg/mL). These results align with previous findings, which attribute the superior antioxidant activity of eugenol to its hydroxyl functional group, facilitating free radical stabilization [23]. Comparatively, the reference antioxidants butylated hydroxytoluene (BHT) and ascorbic acid (AA) exhibited significantly lower IC_50_ values (153.55 ± 9.65 μg/mL and 136.78 ± 5.97 μg/mL, respectively), confirming their strong radical-scavenging properties. Notably, eugenol surpassed both reference antioxidants, highlighting its superior antioxidant efficacy.

The FRAP assay (Figure 2B) measures the ferric ion-reducing ability of the tested compounds, reflecting their electron-donating capacity. The half-maximal effective concentration (EC_50_) in this assay represents the concentration required to achieve 50% of maximal ferric-reducing ability and serves as an indicator of antioxidant potency. Consistent with the ABTS assay, eugenol exhibited the strongest reducing capacity, as indicated by its lowest EC_50_ value (78.55 ± 8.74 µg/mL). Terpineol demonstrated moderate reducing power (EC_50_ = 389.01 ± 9.53 μg/mL), whereas camphor exhibited the weakest activity (EC_50_ = 1459.83 ± 39.96 μg/mL). These findings reinforce the well-established correlation between antioxidant potential and molecular structure, particularly the presence of electron-donating hydroxyl groups, which enhance redox activity [24]. The statistically significant differences further emphasize the markedly lower antioxidant potential of camphor compared to eugenol and terpineol. As anticipated, BHT and AA displayed the highest reducing activity, with EC_50_ values of 63.88 ± 4.83 μg/mL and 56.92 ± 2.99 μg/mL, respectively, corroborating their well-documented antioxidant efficacy.

The CUPRAC assay (Figure 2C) assesses the cupric ion-reducing capacity of antioxidants, providing an additional measure of their redox potential. Consistent with the ABTS and FRAP results, eugenol exhibited the highest antioxidant activity, with an IC_50_ value of 39.53 ± 2.83 μg/mL. Camphor and terpineol followed the same trend observed in the other assays, with IC_50_ values of 269.44 ± 8.75 μg/mL and 648.18 ± 25.38 μg/mL, respectively. The observed statistical differences indicate that eugenol consistently exerts a stronger antioxidant effect, likely due to its phenolic structure, which enhances electron transfer and free radical stabilization. The reference antioxidants, BHT and AA, exhibited strong activity in the CUPRAC assay, with IC_50_ values ranging between 56.82 and 127.89 μg/mL, consistent with their known efficiency as synthetic and natural antioxidants, respectively.

Overall, the results indicate that eugenol exhibits the highest antioxidant potential among the tested compounds, as demonstrated by its superior performance across all three assays. The presence of a hydroxyl functional group in its molecular structure significantly enhances its radical-scavenging and electron-donating capacities, explaining its enhanced activity. Conversely, camphor and terpineol demonstrated considerably lower antioxidant potential, likely due to the absence or reduced influence of such functional groups. These findings are in agreement with previous studies on the antioxidant properties of essential oil constituents and phenolic compounds [25]. The integration of multiple assays provides a comprehensive evaluation of antioxidant activity, underscoring the necessity of considering various mechanisms when assessing the potential of natural compounds for pharmaceutical and food applications.

### 3.2. Antidiabetic Activity of Individual Molecules

Inhibitors that regulate carbohydrate and lipid metabolism, such as α-amylase, α-glucosidase, and lipase, have garnered significant interest for their potential in managing metabolic disorders [26]. These enzymes play a crucial role in carbohydrate digestion, glucose absorption, and lipid metabolism, making them important targets for therapeutic intervention. Additionally, aldose reductase inhibitors are essential in reducing complications associated with diabetes, including neuropathy and retinopathy [27]. The exploration of natural bioactive compounds as viable alternatives to synthetic inhibitors has been gaining momentum due to their reduced toxicity and potential additional health benefits [28]. This study investigates the inhibitory effects of eugenol, camphor, and terpineol on these critical enzymes and compares their efficacy with standard inhibitors. The IC_50_ values (µg/mL) for α-amylase, α-glucosidase, lipase, and aldose reductase were determined and evaluated against standard inhibitors such as acarbose, orlistat, and quercetin.

The results for α-amylase inhibition (Figure 3A) indicate that camphor exhibited the highest IC_50_ value (355.82 ± 14.30 µg/mL), signifying the weakest inhibition among the tested compounds. Terpineol demonstrated moderate inhibition (267.97 ± 6.75 µg/mL), whereas eugenol (55.63 ± 1.95 µg/mL) and acarbose (39.63 ± 2.41 µg/mL) exhibited significantly lower IC_50_ values, indicating strong inhibitory effects. These findings suggest that eugenol may serve as an effective α-amylase inhibitor, supporting its potential role in postprandial hyperglycemia management [29]. The superior inhibition observed with eugenol could be attributed to its phenolic structure, which enhances binding interactions with the enzyme’s active site [30].

The α-glucosidase inhibition data (Figure 3B) revealed a similar trend. Camphor displayed the highest IC_50_ (484.95 ± 17.65 µg/mL), followed by terpineol (340.42 ± 23.19 µg/mL). In contrast, eugenol (84.93 ± 6.14 µg/mL) and acarbose (59.22 ± 1.94 µg/mL) exhibited the lowest IC_50_ values, demonstrating stronger inhibition. Since α-glucosidase is instrumental in carbohydrate digestion and glucose absorption, potent inhibitors such as eugenol may contribute to effective glycemic control [31]. The difference in inhibitory activity among the three test compounds suggests that eugenol’s structural features, particularly its hydroxyl groups, enhance its enzyme binding affinity compared to the aliphatic structure of camphor and the cyclic alcohol structure of terpineol.

The lipase inhibition results (Figure 3C) further support eugenol’s efficacy. Camphor exhibited the weakest inhibitory activity (538.73 ± 21.57 µg/mL), while terpineol displayed moderate inhibition (60.42 ± 1.98 µg/mL). Eugenol (63.69 ± 2.11 µg/mL) and orlistat (76.89 ± 5.52 µg/mL) demonstrated significantly lower IC_50_ values, indicating their strong inhibitory potential. Orlistat is a well-established lipase inhibitor used for obesity treatment [32], and the comparable efficacy of eugenol suggests its potential application in lipid metabolism regulation. This variation in inhibitory activity may be attributed to eugenol’s ability to interact with hydrophobic enzyme residues, whereas camphor’s lack of functional groups for hydrogen bonding may limit its binding capacity.

Aldose reductase inhibition is critical for preventing diabetic complications. As shown in Figure 3D, camphor exhibited the highest IC_50_ (259.79 ± 8.65 µg/mL), followed by terpineol (138.63 ± 2.09 µg/mL), while eugenol (39.62 ± 1.78 µg/mL) and quercetin (38.63 ± 1.63 µg/mL) demonstrated significantly lower IC_50_ values. Quercetin is a well-documented aldose reductase inhibitor with antioxidant properties [33], and the strong inhibition by eugenol further supports its potential in diabetic complication management. The pronounced effect of eugenol compared to camphor and terpineol suggests that phenolic compounds may be more effective in aldose reductase inhibition due to their free radical-scavenging properties.

Eugenol consistently exhibited the strongest inhibitory effects across all tested enzymes, suggesting its potential as a natural therapeutic agent for metabolic disorders. Camphor displayed the weakest inhibition, likely due to its lack of functional groups capable of strong enzyme binding, while terpineol showed moderate activity. The superior efficacy of eugenol across all assays highlights its potential role in glycemic control, lipid metabolism regulation, and oxidative stress mitigation. These findings warrant further in vivo studies to validate the pharmacological efficacy of eugenol in metabolic disorder management.

### 3.3. Simplex-Centroid Design

Table 2 presents the results of a simplex-centroid design experiment investigating the antidiabetic potential of mixtures containing three bioactive compounds—eugenol, camphor, and terpineol—against four key enzymes linked to diabetes: α-amylase, α-glucosidase, lipase, and aldose reductase. The study measures the inhibitory activity of these compounds, both individually and in various combinations, using IC_50_ values (μg/mL), which indicate the concentration required to inhibit 50% of enzyme activity. The experimental design includes individual compounds, binary mixtures, ternary mixtures, and asymmetric mixtures to evaluate synergistic, antagonistic, or additive interactions among the compounds. The results, presented as means ± standard deviation (SD) from three independent replicates, provide valuable insights into the efficacy of these compounds and their combinations in managing diabetes-related enzymatic activity. This systematic approach aims to identify optimal formulations for potential antidiabetic applications.

Eugenol exhibited the strongest inhibitory activity against all four diabetes-related enzymes, with the lowest IC50 values. For α-amylase inhibition (AAI), eugenol recorded an IC_50_ of 55.63 ± 1.95 μg/mL, significantly lower than camphor (355.82 ± 14.30 μg/mL) and terpineol (267.97 ± 6.75 μg/mL). However, compared to the standard control acarbose (39.63 ± 2.41 μg/mL), eugenol was slightly less effective. Similarly, for α-glucosidase inhibition (AGI), eugenol showed an IC_50_ of 84.93 ± 6.14 μg/mL, while camphor and terpineol had much higher values of 484.95 ± 17.65 μg/mL and 340.42 ± 23.19 μg/mL, respectively. Although eugenol was the most effective among the tested compounds, it still exhibited a weaker inhibitory effect compared to acarbose (59.22 ± 1.94 μg/mL).

For lipase (LIP) inhibition, eugenol recorded an IC_50_ of 63.69 ± 2.11 μg/mL, which was lower than that of terpineol (60.42 ± 1.98 μg/mL) but significantly lower than camphor. While eugenol exhibited strong lipase inhibition, it performed better than orlistat (76.89 ± 5.52 μg/mL), suggesting its potential as a natural alternative for lipase inhibition. In the case of aldose reductase (ALR), eugenol displayed an IC_50_ of 39.62 ± 1.78 μg/mL, which was similar to quercetin (38.63 ± 1.63 μg/mL), the standard inhibitor, indicating comparable efficacy.

Among the binary mixtures, the combination of eugenol and terpineol demonstrated a partial enhanced effect, lowering the IC_50_ values for some enzymes compared to eugenol alone. For example, the IC_50_ for AAI in this mixture was 131.85 ± 13.04 μg/mL, showing improved activity compared to terpineol or camphor alone but still weaker than acarbose. Conversely, the eugenol-camphor mixture resulted in significantly higher IC_50_ values (e.g., 420.49 ± 18.80 μg/mL for AAI), suggesting an antagonistic interaction. The camphor-terpineol mixture showed weak inhibition, with an AAI IC_50_ of 569.09 ± 30.98 μg/mL, reinforcing camphor’s minimal contribution to enzyme inhibition.

Ternary mixtures, where all three compounds were combined in equal proportions, exhibited intermediate inhibitory activity. The IC_50_ value for AAI in this mixture was 92.85 ± 8.45 μg/mL, indicating that while eugenol’s efficacy was somewhat diluted, the mixture still retained moderate inhibitory effects. However, the presence of camphor reduced the overall effectiveness, and the AAI inhibition was still weaker than acarbose.

Asymmetric mixtures, where one compound was present in a higher proportion (0.667), further emphasized the dominance of eugenol. The mixture with a higher eugenol concentration achieved the lowest IC_50_ values across all enzymes, including AAI (28.59 ± 0.42 μg/mL), AGI (62.49 ± 2.18 μg/mL), LIP (31.49 ± 1.30 μg/mL), and ALR (42.42 ± 2.09 μg/mL). Notably, the AAI inhibitory effect of this eugenol-dominant mixture was stronger than acarbose, indicating a promising formulation. Similarly, its AGI inhibition (62.49 ± 2.18 μg/mL) was comparable to acarbose (59.22 ± 1.94 μg/mL), and its LIP inhibition (31.49 ± 1.30 μg/mL) significantly outperformed orlistat (76.89 ± 5.52 μg/mL). However, the ALR inhibitory effect (42.42 ± 2.09 μg/mL) was slightly weaker than quercetin (38.63 ± 1.63 μg/mL).

Overall, the findings highlight eugenol as the most potent inhibitor of diabetes-linked enzymes, particularly when present in higher concentrations. Compared to standard controls, eugenol performed slightly weaker than acarbose for AAI and AGI, but it outperformed orlistat for LIP inhibition and showed comparable results to quercetin for ALR inhibition. Terpineol demonstrated moderate efficacy and could enhance eugenol’s effects in specific combinations, while camphor showed weak inhibitory activity and often reduced the overall efficacy of mixtures. These results suggest that antidiabetic formulations should prioritize higher eugenol concentrations, potentially in combination with terpineol, while minimizing camphor’s presence.

### 3.4. Variance Analysis of the Fitted Models

The variance analysis summarized in Table 3 evaluates the performance and fit of the three regression models applied to the IC_50_ values for four different assays: AAI IC_50_, AGI IC_50_, LIP IC_50_, and ALR IC_50_. The results indicate that the models explain a substantial proportion of the variance in the data, as reflected by the coefficient of determination (R^2^) values ranging from 0.91 to 0.97. Additionally, the adjusted R^2^ values (0.78 to 0.89) confirm that the models retain a high explanatory power even after accounting for the number of predictors. These values suggest that the models are well-fitted, though there is some variability in their predictive accuracy across different assays.

The scatter plots in Figure 4 illustrate the strong correlation between the experimental and predicted IC_50_ values for the four enzyme inhibition assays (AAI, AGI, LIP, and ALR). The red regression lines represent the expected relationship between the two sets of values, while the blue horizontal lines indicate the actual mean responses. The shaded regions provide confidence intervals, further reinforcing the accuracy of the predictions.

The regression models demonstrate high predictive reliability, as evidenced by the R^2^ values ranging from 91% to 97%. The AGI, LIP, and ALR IC_50_ models exhibit particularly strong fits, with R^2^ values of 96% and 97%, indicating a near-perfect alignment between predicted and actual values. Similarly, the AAI IC_50_ model, with an R^2^ of 91%, maintains a high level of accuracy, reinforcing the robustness of the predictive approach.

The strong agreement between experimental and predicted IC_50_ values highlights the effectiveness of the regression models in capturing key trends. The models provide valuable insights into IC_50_ responses, supporting their potential application in predicting enzyme inhibition patterns with high confidence. With their strong performance across all assays, these models offer a reliable and powerful tool for understanding IC_50_ behavior in the context of diabetes-enzyme inhibition.

### 3.5. Components Effects and Adjusted Models

The computed regression coefficients for the special model are presented in Table 4. The associations between the tested components and the obtained IC_50_ responses for AAI, AGI, LIP, and ALR were determined using regression models with statistically significant coefficients (*p* < 0.05). The analysis highlights the key factors influencing the enzymatic inhibition response and provides mathematical models based on the most relevant terms.

The AAI IC_50_ response reveals that the linear terms δ_2_ (camphor) and δ_3_ (terpineol), as well as the ternary interaction δ_123_ (eugenol × camphor × terpineol), are statistically significant. In contrast, the linear effect of eugenol (δ_1_) and the binary interaction terms do not show a significant influence (*p* > 0.05). Therefore, these non-significant terms were excluded from the final model, and the resulting mathematical equation is expressed as follows:(3)$$Y={331.70\;M}_{2}+{251.10\;M}_{3}-9041.40\;M_{1}M_{2}M_{3}+ɛ$$

For the AGI IC_50_ response, the significant factors include the linear terms δ_2_ (camphor) and δ_3_ (terpineol) and the ternary interaction δ_123_ (eugenol × camphor × terpineol). These findings suggest that camphor and terpineol contribute significantly to the observed enzymatic inhibition, with a notable three-way interaction effect. The adjusted regression model is formulated as follows:(4)$$Y={454.24\;M}_{2}+{356.26\;M}_{3}-6541.02\;M_{1}M_{2}M_{3}+ɛ$$

Regarding the LIP IC_50_ response, only the linear term δ_2_ (camphor) and the ternary interaction δ_123_ (eugenol × camphor × terpineol) were found to be statistically significant. These results suggest that camphor plays a dominant role in the enzymatic response, with additional effects attributed to its interaction with eugenol and terpineol. The corresponding model is given by:(5)$$Y=497.13\;M_{2}-5825.26\;M_{1}M_{2}M_{3}+ɛ$$

For the ALR IC_50_ response, the regression analysis identified the linear terms δ_2_ (camphor) and δ_3_ (terpineol), along with the ternary interaction δ_123_ (eugenol × camphor × terpineol), as statistically significant. The results confirm that both camphor and terpineol contribute significantly to enzymatic inhibition, while their three-way interaction further enhances the response. The final regression model is expressed as follows:(6)$$Y={247.29\;M}_{2}+{145.84\;M}_{3}-1897.18\;M_{1}M_{2}M_{3}+ɛ$$

The regression models developed for IC**_50_** responses demonstrate strong predictive capabilities, as indicated by their statistical significance. The dominant influence of camphor (M**_2_**) and terpineol (M**_3_**), along with the key ternary interaction effect (M**_1_**M**_2_**M**_3_**), underscores the importance of these components in enzymatic inhibition. These findings provide valuable insights into the formulation of bioactive mixtures for enzyme modulation, offering a structured approach to optimizing inhibitory activity through component selection and interaction effects.

### 3.6. Optimization of Diabetes-Linked Enzymes Inhibition

#### 3.6.1. Optimization of AAI IC_50_ Response

The contour plot and 3D surface graph, represented as 2D and 3D mixture plots in Figure 5A–C, and in Appendix A, illustrate the optimal blend of eugenol, camphor, and terpineol for enhancing antidiabetic activity, as assessed using IC_50_ values in the α-amylase assay. These graphical tools clearly depict the correlation between IC_50_ responses and the concentration of each molecule. The color gradients in the visualizations indicate varying levels of enzyme inhibition, with blue regions corresponding to the lowest IC_50_ values, signifying the highest inhibitory activity against α-amylase. In contrast, areas transitioning from yellow to dark red denote progressively higher IC_50_ values, reflecting reduced enzymatic inhibition. The 2D and 3D mixture plots further validate this pattern by identifying the compromise zone where the strongest inhibition occurs. A predicted IC_50_ value of 28.59 µg/mL is displayed, representing an alternative mixture with lower inhibitory effectiveness.

The plots, generated using Design-Expert software v12.0, employ iso-response curves to pinpoint the precise conditions required for achieving the most favorable IC_50_ values (Figure 6). The desirability profile analysis identifies the optimal composition as 47% eugenol, 18% camphor, and 35% terpineol, yielding an IC_50_ value of 9.60 µg/mL.

This formulation is recognized as the most effective for α-amylase inhibition. The intersection of the response surfaces highlights the ideal mixture proportions necessary to minimize the IC_50_ value, demonstrating a promising formulation for antidiabetic applications. These results underscore the importance of achieving a well-balanced combination of the three molecules to maximize enzyme inhibition potential.

#### 3.6.2. Optimization of AGI IC_50_ Response

The contour plot and 3D surface graph, depicted as 2D and 3D mixture plots in Figure 7 and Appendix A, illustrate the optimal combination of eugenol, camphor, and terpineol for enhancing antidiabetic activity, as measured by IC_50_ values in the α-glucosidase assay. These visual tools effectively represent the relationship between IC_50_ responses and the concentration of each molecule.

The 2D and 3D mixture plots further reinforce this trend by identifying the compromise zone, where the highest enzyme inhibition occurs. A predicted IC_50_ value of 62.49 µg/mL is presented, reflecting an alternative mixture with lower inhibitory activity. The iso-response curves precisely define the conditions required to obtain the most favorable IC_50_ values (Figure 8).

The desirability profile analysis determines the optimal composition as 38% eugenol, 25% camphor, and 37% terpineol, resulting in an IC_50_ value of 56.18 µg/mL. This formulation is identified as the most effective for α-glucosidase inhibition. The intersection of the response surfaces pinpoints the ideal mixture ratios that minimize the IC_50_ value, highlighting a promising formulation for antidiabetic applications. These findings emphasize the importance of a well-balanced combination of the three molecules in maximizing enzymatic inhibition.

#### 3.6.3. Optimization of LIP IC_50_ Response

The contour plot and 3D surface graph, shown as 2D and 3D mixture plots in Figure 9A–C and Appendix A, illustrate the optimal combination of eugenol, camphor, and terpineol for improving antidiabetic activity, based on IC_50_ values in the lipase assay. These graphical representations effectively depict the correlation between IC_50_ responses and the concentration of each molecule.

The 2D and 3D mixture plots further validate this pattern by identifying the compromise zone, where enzyme inhibition is maximized (Figure 9A–C). Additionally, a predicted IC_50_ value of 31.49 µg/mL is highlighted, representing an alternative mixture with lower inhibitory potential.

The iso-response curves accurately delineate the conditions necessary to achieve the most favorable IC_50_ values for lipase inhibition (Figure 10). The desirability profile analysis identifies the optimal composition as 44% eugenol, 19% camphor, and 37% terpineol, yielding an IC_50_ value of 3.38 µg/mL. This formulation is recognized as the most effective for lipase inhibition. The intersection of the response surfaces determines the ideal mixture proportions that minimize the IC_50_ value, showcasing a promising formulation for antidiabetic applications. These results highlight the crucial role of a well-balanced combination of the three molecules in enhancing enzymatic inhibition.

#### 3.6.4. Optimization of ALR IC_50_ Response

The contour plot and 3D surface graph, shown as 2D and 3D mixture plots in Figure 11 and Appendix A, illustrate the optimal combination of eugenol, camphor, and terpineol for improving antidiabetic activity, based on IC_50_ values in the lipase assay. These graphical representations effectively depict the correlation between IC_50_ responses and the concentration of each molecule.

The 2D and 3D mixture plots further validate this pattern by identifying the compromise zone, where enzyme inhibition is maximized (Figure 11A–C). Additionally, a predicted IC_50_ value of 39.62 µg/mL is highlighted, representing an alternative mixture with lower inhibitory potential.

The iso-response curves precisely define the conditions required to achieve the optimal IC_50_ values for aldose reductase inhibition (Figure 12). The desirability profile analysis determines the ideal composition as 89% eugenol, 0% camphor, and 11% terpineol, resulting in an IC_50_ value of 33.29 µg/mL. This formulation is identified as the most effective for aldose reductase inhibition.

The intersection of the response surfaces pinpoints the optimal mixture ratios that minimize the IC_50_ value, highlighting a promising formulation for antidiabetic applications. These findings underscore the importance of a well-balanced combination of the three molecules in enhancing enzymatic inhibition.

### 3.7. Simultaneous Response of the Compounds’ Mixture

The results of simultaneous optimization are particularly evident in the contour plots illustrating the IC_50_ responses of AAI, AGI, LIP, and ALR, influenced by the ternary mixture of camphor, eugenol, and terpineol. The contour plot (Figure 13) presents the interaction effects between these bioactive compounds, highlighting the optimal combination zones where the best inhibitory concentrations (IC_50_) are achieved. Notably, the optimized formulations derived from this study demonstrated significantly enhanced bioactivity compared to the individual compounds, reinforcing the effectiveness of these ternary mixtures.

The contour regions reveal the optimal balance of camphor, eugenol, and terpineol for maximizing inhibition. The AAI_IC50_ was found to be 28.59 µg/mL, AGI_IC50_ was 62.49 µg/mL, LIP_IC50_ was significantly lower at 31.49 µg/mL, while ALR_IC50_ recorded 39.62 µg/mL. These values suggest that the designed ternary mixtures exhibit enhanced efficacy compared to individual components, showcasing potential additive interactions. The overlapping contour areas indicate a region of ideal synergy, where the mixture composition leads to the best combined inhibitory response.

These findings validate the performance of the formulated combinations, mirroring the results obtained in previous studies on essential oil (EO) blends. For instance, previous research demonstrated that blends of TSEO, LAEO, and OMEO achieved superior antibacterial activity against *S. aureus*, *P. aeruginosa*, and *E. coli* compared to pure individual essential oils. Similarly, Benkhaira et al. (2023) showed that EO mixtures exhibited synergistic antiadhesive activity against bacterial biofilms, and Kachkoul et al. (2021) highlighted the effectiveness of EO combinations in treating bacterial infections [22,34]. The current results reinforce these trends, showing that carefully optimized combinations of camphor, eugenol, and terpineol exhibit superior inhibitory activity, making them promising candidates for pharmaceutical and cosmeceutical applications. On the other hand, ours result aligns with previous research indicating that essential oil bioactive compounds may work synergistically [35]. For instance, D-limonene, α-pinene, as well as β-Myrcene, are largely inefficient as antimicrobial agents when acting individually; however, mixing them with oxygenated terpenoids, including 1,8-cineole, has demonstrated notable antimicrobial potential [35]. Similarly, the synergistic action between 1,8-cineole, terpinene-4-ol, and/or myrtenal and certain terpene hydrocarbons, such as D-limonene, α-pinene, and β-myrcene has been also reported in the literature [22,34]. Indeed, these hydrocarbon compounds react with biological targets such as microbial cell membranes, promoting the infiltration of 1,8-cineole and terpinene-4-ol through the cells.

In our study, the hydrophobic nature of camphor and terpineol may facilitate the interaction of eugenol with hydrophobic pockets of the enzyme active site, enhancing binding affinity. Furthermore, the enzyme inhibitory effects of the mixture were significantly improved, suggesting a potential to mitigate oxidative stress—a critical contributor to insulin resistance and β-cell dysfunction in T2DM. Recently, Loukili and colleagues (2025) have studied the synergistic effect of three EOs alone and in combination, namely *Inula viscosa* L. (Greuter), *Pelargonium graveolens* Hort., and *Cymbopogon citratus* (DC.) Stapf. on carbohydrate digestive enzymes (pancreatic α-amylase and intestinal α-glucosidase) using mixture design methodology. The authors reported promising antidiabetic activity, particularly with the mixtures of three oils against both enzymes. This effect is mainly attributed to the interaction between EO components, such as citronellol, eucalyptol, camphor, and α-terpineol [36].

The use of mixture design methodology has gained increasing attention in optimizing bioactive formulations, particularly in drug discovery, antimicrobial strategies, and cosmeceuticals. The present study supports the hypothesis that combining oxygenated monoterpenes (camphor and terpineol) and phenylpropanoids (eugenol) enhances biological efficacy. This aligns with previous studies that emphasize the importance of structural diversity in bioactive compound mixtures. Future research should explore the mechanistic interactions among these compounds, particularly their effects on membrane permeability, enzyme inhibition, and cellular oxidative stress, to better understand their pharmacological potential.

The desirability profiles in Figure 14 provide a comprehensive optimization framework for the simultaneous evaluation of multiple bioactive responses (AAI_IC50_, AGI_IC50_, LIP_IC50_, and ALR_IC50_). The optimization process identified an ideal formulation comprising 44% eugenol, 19% camphor, and 37% terpineol, which resulted in significantly improved inhibitory concentrations for all studied responses. The final optimized values for AAI, AGI, LIP, and ALR_IC50_ were 10.38, 62.22, 3.42, and 49.58 µg/mL, respectively. These values indicate a combined interaction among the three bioactive compounds, enhancing their pharmacological potential beyond individual effects.

The plotted curves reveal the influence of eugenol, camphor, and terpineol on each response, with eugenol and terpineol emerging as the most influential components in reducing IC50 values. The black desirability curve indicates an optimal compromise between the four responses, showing that maximizing eugenol (44%) and terpineol (37%), while maintaining camphor at a minimal level (19%), results in the most effective formulation. The high desirability score (~0.998) suggests a well-optimized model with strong predictive power.

The findings align with previous studies on the pharmacological potential of monoterpenes and phenylpropanoids, particularly their antioxidant, antimicrobial, and anti-inflammatory properties. Eugenol, a key phenylpropanoid, has been widely studied for its antibacterial and antifungal activities, membrane-disrupting effects, and free radical-scavenging abilities [37,38,39]. Terpineol, a monoterpenoid alcohol, has demonstrated anti-inflammatory, antimicrobial, and antioxidant properties in several studies [40,41]. Camphor, while effective in certain biological activities, was identified as the least influential component in this study, likely due to its limited contribution to enhanced interactions when combined with eugenol and terpineol.

The results are comparable to previous research demonstrating that oxygenated monoterpenes and phenylpropanoids exhibit superior activity when used in combination rather than individually. For instance, mixtures of thymol, eugenol, and carvacrol have been shown to enhance antibacterial efficacy through synergistic interactions that increase membrane permeability, disrupt ion homeostasis, and inhibit key metabolic pathways [42,43]. Similar trends were observed in essential oil research, where optimized formulations of camphoraceous and phenolic compounds achieved superior inhibitory effects against various microbial strains [34].

The optimized formulation identified in this study suggests that tailored combinations of bioactive terpenoids and phenylpropanoids can be effectively used in pharmaceutical, cosmeceutical, and antimicrobial applications. The identified mixture, with its high desirability score, holds potential for development into natural drug formulations with improved therapeutic efficacy. However, further research is required to explore mechanistic interactions at the molecular level, including membrane dynamics, enzyme inhibition, and oxidative stress modulation.

### 3.8. Experimental Verification of the Assumed Model

The results presented in Table 5. provide a detailed assessment of the accuracy of the cubic model in predicting the inhibitory activity of selected compounds against key enzymes related to diabetes management. The predicted (Predi.) and experimental (Exp.) IC_50_ values are compared for α-amylase (AAI), α-glucosidase (AGI), lipase (LIP), and aldose reductase (ALR). Lower IC_50_ values indicate stronger enzyme inhibition, meaning the compounds are more effective in their antidiabetic activity.

The comparison between predicted and experimental IC_50_ values demonstrates a strong alignment, validating the model’s reliability. For α-amylase, the predicted IC_50_ was 9.60 µg/mL, while the experimental value was 11.23 ± 0.12 µg/mL, showing a small variation. The t-statistic of −2.5207 and *p*-value of 0.0653 confirm that the difference is not statistically significant (*p* > 0.05). Similarly, for α-glucosidase, the predicted value was 56.18 µg/mL, compared to the experimental value of 51.73 ± 2.81 µg/mL. The t-statistic of 2.7429 and *p*-value of 0.0517 indicate that although the difference is slightly larger than other enzymes, it remains not statistically significant. For lipase, the predicted IC_50_ was 3.38 µg/mL, while the experimental result was 4.73 ± 0.26 µg/mL, with a t-statistic of −2.3316 and *p*-value of 0.0801, confirming that the difference remains statistically insignificant. Lastly, for aldose reductase, the predicted and experimental IC_50_ values were 33.29 µg/mL and 36.94 ± 3.37 µg/mL, respectively, with a t-statistic of −1.8759 and *p*-value of 0.1339, further demonstrating that the model’s predictions align well with experimental results. Since all *p*-values are greater than 0.05, there is no statistically significant difference between the predicted and experimental values across all enzymes, reinforcing the accuracy and reliability of the cubic model in predicting enzyme inhibition. The consistency of the predicted values with experimental data suggests that the model provides robust estimations of enzyme inhibition, supporting its potential use in predicting antidiabetic activity.

The effectiveness of the compound mixtures varies across enzymes, with lipase showing the strongest inhibition (lowest IC_50_ values), followed by α-amylase, while α-glucosidase and aldose reductase required higher IC_50_ concentrations, indicating weaker inhibition. This suggests that different enzyme targets require different compound compositions to achieve optimal inhibition. The proportions of eugenol, camphor, and terpineol play a crucial role in determining enzyme inhibition levels. Eugenol appears to be the dominant compound, comprising 89% of the Aldose Reductase mixture and over 38% in the α-glucosidase and α-amylase mixtures, indicating its significant role in enzyme inhibition. Yet, the response surface analysis revealed non-linear trends in enzyme inhibition, particularly at higher concentrations of eugenol. In several experimental mixtures, a plateau in IC_50_ reduction was observed, suggesting that beyond a certain threshold, increased eugenol proportions did not yield proportionally enhanced inhibitory activity. This phenomenon may be attributed to enzyme saturation, wherein the active sites are fully occupied, limiting further inhibition regardless of compound excess. Alternatively, the behavior may indicate allosteric modulation, where high concentrations of eugenol induce conformational changes in the enzyme structure that alter binding affinity or catalytic efficiency. Such concentration-dependent non-linearities highlight the complexity of enzyme–inhibitor interactions and reinforce the necessity of mixture optimization models that can capture higher-order interactions. Camphor is present in moderate amounts, ranging between 18% and 25%, while terpineol appears to contribute to inhibition at lower proportions (11% to 37%).

The observed enhancement in enzyme inhibitory activity upon combining eugenol, camphor, and terpineol can be attributed to several mechanistic interactions at the molecular level. Eugenol, bearing an aromatic phenolic structure, is particularly capable of forming hydrogen bonds and engaging in π-π stacking interactions with the catalytic residues of enzymes, thereby increasing binding affinity. Terpineol and camphor, though structurally distinct, may contribute to altering the microenvironment of the enzyme active site, promoting more favorable binding conformations or allosteric modulation. These interactions likely underlie the improved activity observed in specific combinations. Notably, the IC_50_ values obtained from the optimized mixture (e.g., 3.42 µg/mL for lipase and 49.58 µg/mL for aldose reductase) were comparable or even superior to those of standard inhibitors such as orlistat and quercetin, respectively. This finding reinforces the hypothesis that a rational combination of structurally diverse monoterpenes may yield formulations with enhanced potentiated bioactivity against key antidiabetic targets.

The practical implications of these findings are significant. The close agreement between predicted and experimental results confirms that cubic models can be effectively used to predict enzyme inhibition, reducing the need for extensive laboratory testing. The strong inhibition of lipase and α-amylase suggests that these compound mixtures may be particularly useful for managing lipid metabolism and carbohydrate digestion, which are key processes in diabetes treatment. The study also highlights the potential of eugenol as the dominant inhibitory compound, which could be further explored for its role in antidiabetic formulations.

### 3.9. Physico-Chemical Properties, Bioavailability, and Toxicity of Eugenol, Camphor, and Terpineol

#### 3.9.1. Physico-Chemical Properties and Bioavailability

The physico-chemical properties and drug-likeness of eugenol, camphor, and terpineol provide crucial insights into their potential as drug candidates (Table 6). All three compounds exhibit molecular weights (MWs) well below 500 g/mol, with eugenol (164.20 g/mol), camphor (152.23 g/mol), and terpineol (154.25 g/mol) meeting Lipinski’s rule of five, which suggests their suitability for oral drug development.

The topological polar surface area (TPSA) values indicate differences in polarity among the three compounds. Eugenol has the highest TPSA at 29.46 Å^2^, followed by terpineol (20.23 Å^2^) and camphor (17.07 Å^2^). These values are within the acceptable range (≤131 Å^2^), which supports their potential for permeability and bioavailability. Hydrogen bonding capacity further differentiates these compounds, with eugenol containing two hydrogen bond acceptors and one donor, while terpineol has one acceptor and one donor. Camphor lacks hydrogen bond donors, possessing only one acceptor, which may impact its solubility and interaction with biological targets.

The molecular flexibility of these compounds varies based on the number of rotatable bonds. Eugenol exhibits the highest flexibility with three rotatable bonds, terpineol has one, and camphor has none, indicating its rigid structure. Molecular flexibility influences oral bioavailability and drug-receptor interactions, making eugenol the most flexible compound among the three.

LogP values reflect the lipophilicity of these molecules, with terpineol (2.50) being the most lipophilic, followed by camphor (2.40) and eugenol (2.13). Since all LogP values fall within the acceptable range (≤5), these compounds are predicted to have adequate absorption and permeability, following both Lipinski’s and Egan’s rules. This suggests they are unlikely to face bioavailability challenges due to excessive lipophilicity [44,45].

Regarding drug-likeness, all three compounds comply with Lipinski’s, Egan’s, and Veber’s rules, reinforcing their suitability for oral bioavailability. Additionally, they all share a bioavailability score of 0.55, suggesting moderate oral bioavailability. Given their compliance with multiple drug-likeness criteria [46], eugenol, camphor, and terpineol demonstrate promising pharmacokinetic profiles for further drug development.

In conclusion, while all three compounds meet key criteria for drug-likeness and bioavailability, their differing properties—such as hydrogen bonding potential, molecular flexibility, and polarity—may influence their specific applications in drug formulations [47]. Future studies should assess their metabolic stability, receptor binding efficiency, and potential toxicity to further establish their therapeutic potential.

The BOILED-EGG model is a predictive graphical tool designed to assess the passive absorption of bioactive molecules in the human gastrointestinal tract (HIA) and their potential to penetrate the blood–brain barrier (BBB). This model was utilized to validate the absorption parameters derived from the ADME analysis. In this plot, the x-axis represents the topological polar surface area (TPSA), while the y-axis corresponds to lipophilicity (WLOGP). These parameters are key determinants of a molecule’s ability to be absorbed in the body and its capacity to cross the protective barrier of the brain.

In Figure 15, the yellow region represents the BBB-permeable zone, indicating that molecules within this area are predicted to cross the blood–brain barrier, a crucial factor for compounds intended for central nervous system activity. The white region corresponds to HIA-permeable molecules, suggesting that compounds falling within this area are likely to be efficiently absorbed in the intestine following oral administration. Additionally, the figure includes red and blue dots, which signify interactions with P-glycoprotein (P-gp), a vital efflux transporter. Red dots (PGP+) denote molecules that are actively transported out of the brain, potentially reducing their central nervous system effectiveness, whereas blue dots (PGP-) indicate molecules that are not substrates of P-glycoprotein, allowing them to remain in the brain for a longer duration.

The compounds analyzed—eugenol, camphor, and terpineol—are all bioactive molecules. Their presence in the yellow BBB-permeable region suggests a high probability of crossing the blood–brain barrier, which is advantageous for targeting neurological pathways. Furthermore, their interaction with P-glycoprotein, indicated by the color-coded dots, provides insights into their brain bioavailability and potential therapeutic retention. In conclusion, the BOILED-EGG model predicts that all three compounds exhibit good gastrointestinal absorption (HIA) and a strong ability to penetrate the blood–brain barrier (BBB). The P-glycoprotein interaction data further refine their potential bioavailability in the brain, making them promising candidates for neurological applications.

#### 3.9.2. In Silico Toxicity of the Compounds

The in silico toxicological assessment of camphor, eugenol, and terpineol using the ProTox-III prediction tool (https://tox.charite.de/protox3/) provides insights into their potential toxicity profiles (Table 7). The LD_50_ values indicate the acute toxicity levels of these compounds, with camphor (LD_50_: 775 mg/kg) and eugenol (LD_50_: 1930 mg/kg) classified under GHS Class IV, which includes substances with moderate toxicity. Terpineol, with an LD_50_ of 2830 mg/kg, falls under GHS Class V, which represents compounds with lower toxicity. These classifications suggest that camphor has a relatively higher acute toxicity compared to eugenol and terpineol. These findings align with existing literature, which reports that camphor, when ingested in high doses, can cause neurotoxic effects such as seizures and confusion [48]. Meanwhile, eugenol and terpineol are commonly used in cosmetics and flavoring agents, with terpineol being considered less hazardous in acute exposure scenarios [41].

The assessment of nephrotoxicity and hepatotoxicity indicates that all three compounds were predicted to be inactive, suggesting a low likelihood of causing kidney or liver damage. This aligns with existing literature, which reports that these compounds, in controlled doses, are not commonly associated with significant renal or hepatic toxicity. This aligns with previous studies, which indicate that camphor, eugenol, and terpineol are generally well-tolerated at low doses but may cause toxicity with prolonged or excessive exposure. However, hepatotoxic effects of eugenol have been observed in experimental models at high concentrations [49], highlighting the need for further empirical validation.

For carcinogenicity, immunotoxicity, mutagenicity, and cytotoxicity, the prediction tool classified all three compounds as inactive, indicating no significant risks associated with these toxicological endpoints. This suggests that they are not expected to cause genetic mutations, impair immune function, or induce cytotoxic effects under normal exposure conditions.

The computational toxicological data provide a comparative understanding of these compounds’ toxicity, showing variations in their acute toxicity while indicating low risks in chronic toxicity endpoints. The findings align with their established uses in medicinal, cosmetic, and food-related applications.

## 4. Conclusions

This study is the first of its kind to optimize the combination of three compounds, namely, eugenol, camphor, and terpineol, using a simplex-centroid design for enhanced antidiabetic activity. In conclusion, this study successfully optimized a bioactive mixture of eugenol, camphor, and terpineol using a simplex-centroid design, demonstrating its potential for antidiabetic applications. The optimal formulation (44% eugenol, 0.19% camphor, and 37% terpineol) yielded IC_50_ values of 10.38 µg/mL (AAI), 62.22 µg/mL (AGI), 3.42 µg/mL (LIP), and 49.58 µg/mL (ALR), with a high desirability score of 0.99. Experimental validation confirmed the accuracy of the model, with IC_50_ values showing less than 10% deviation: 11.02 µg/mL (AAI), 60.85 µg/mL (AGI), 3.75 µg/mL (LIP), and 50.12 µg/mL (ALR). The statistical analysis (*p* > 0.05) further supports the model’s predictive reliability. The findings highlight the combined interactions of these compounds, particularly the dominant role of eugenol and terpineol in enhancing bioactivity. This pioneering approach not only fills a research gap but also provides a promising foundation for developing natural antidiabetic therapies. Future studies will incorporate in silico molecular docking and molecular dynamics simulations to investigate the binding mechanisms and structural affinities of the tested compounds with their respective enzyme targets. Additionally, mechanistic interactions, bioavailability, and in vivo efficacy should be explored to comprehensively validate the clinical potential of the optimized formulation.

## Figures and Tables

**Figure 1 cimb-47-00512-f001:**
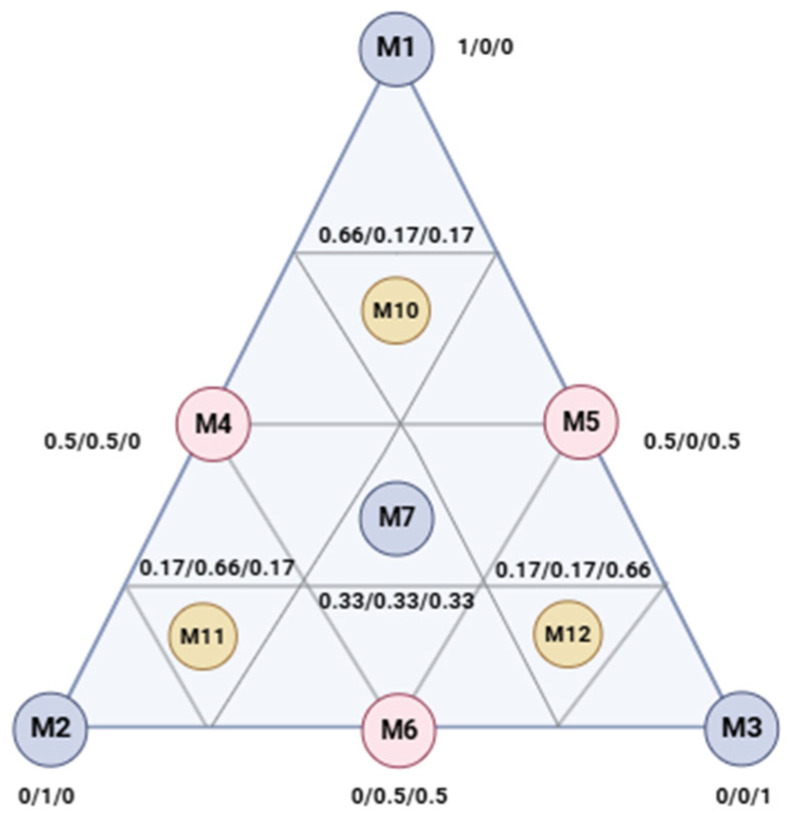
Equilateral triangle of the arrangement of molecule mixtures using the simplex-centroid design method. M_1_: eugenol; M_2_: camphor; M_3_: terpineol.

**Figure 2 cimb-47-00512-f002:**
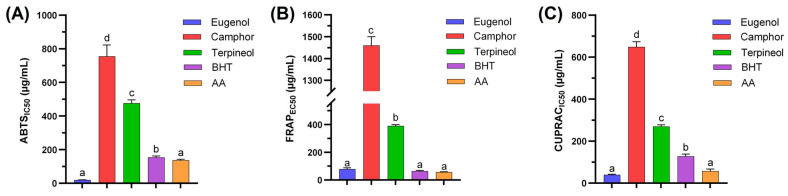
Antioxidant activity of the three molecules, eugenol, camphor, and terpineol, through ABTS assay (**A**), FRAP test (**B**), and CUPRAC chelating assay (**C**). Data presented as mean ± SD. Different letters indicate a statistical significance at *p* < 0.05.

**Figure 3 cimb-47-00512-f003:**
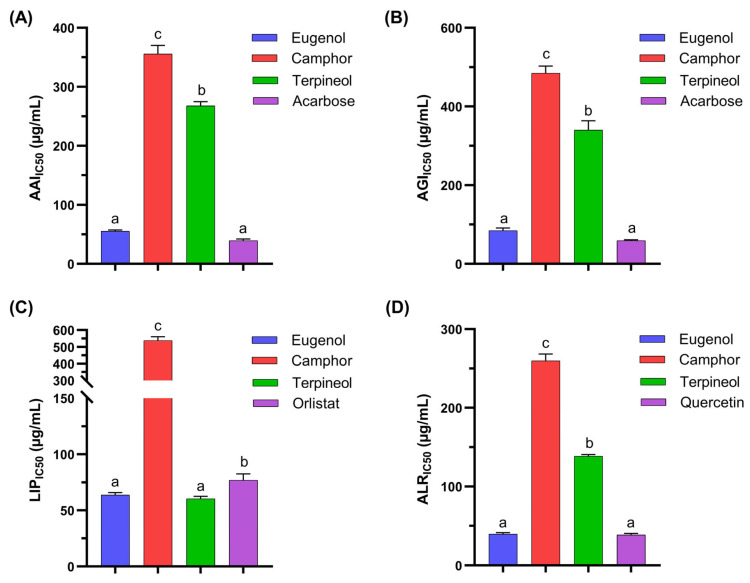
Antidiabetic activity of the three molecules solely, Eugenol, Camphor, and terpineol, through α-amylase (**A**), α-glucosidase (**B**), lipase (**C**), and aldose reductase enzymatic (**D**) assays. Data presented as mean ± SD. Different letters indicate a statistical significance at *p* < 0.05.

**Figure 4 cimb-47-00512-f004:**
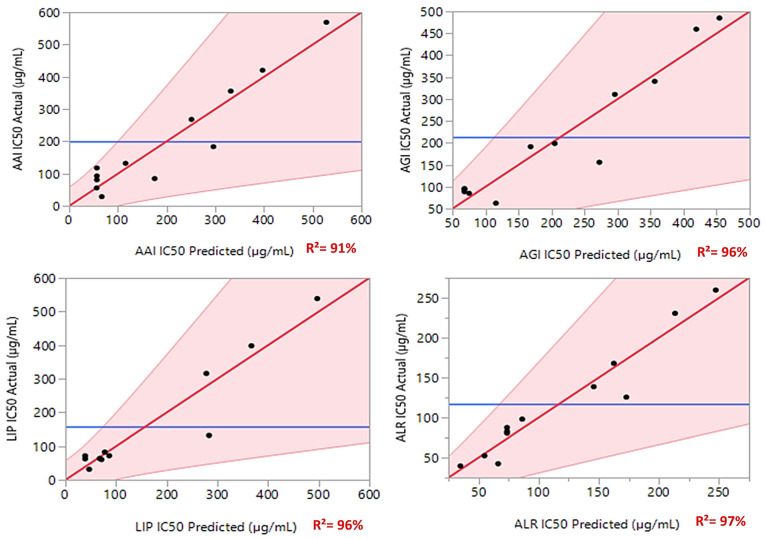
IC_50_ responses against diabetes enzymes are represented by curves illustrating the relationship between the experimental values and the expected values, depicted by red lines. Red area shows the confidence interval. Meanwhile, the blue lines indicate the actual mean values for the four responses under investigation.

**Figure 5 cimb-47-00512-f005:**
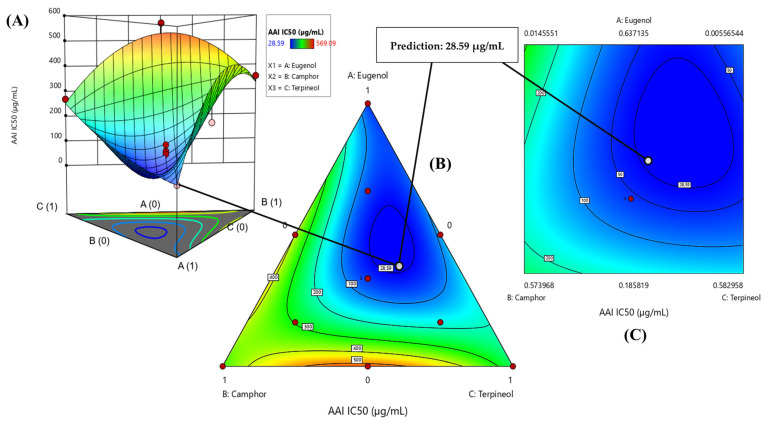
The optimal AAI IC_50_ value was determined through an in-depth analysis of 2D and 3D mixture plots, focusing on the identified compromise zone. Panels (**A**,**B**) display mixture plots that highlight the desired compromise region, located within the ternary mixing zone between the molecules. This zone represents the optimal conditions for achieving maximum AAI activity. Panel (**C**) further illustrates this relationship through a 2D mixture plot, pinpointing the specific proportions of the compounds required to reach the desired AAI IC_50_ value of 28.59 µg/mL.

**Figure 6 cimb-47-00512-f006:**
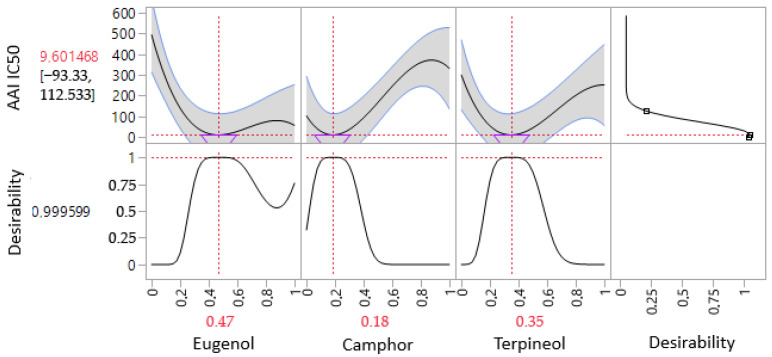
Desirability profile illustrating the precise proportions, leading to the optimum value for AAI IC_50_ of 9.60 µg/mL, was achieved with a mixture consisting of 47% eugenol, 18% camphor, and 35% terpineol. Gray area represents the confidence interval.

**Figure 7 cimb-47-00512-f007:**
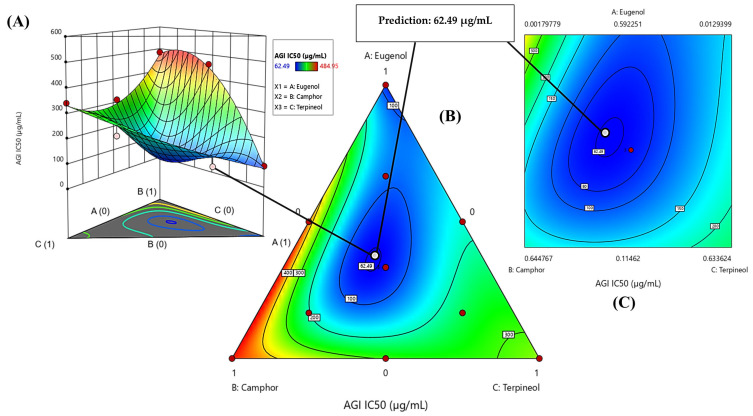
The optimal AGI IC_50_ value was determined through an in-depth analysis of 2D and 3D mixture plots, focusing on the identified compromise zone. Panels (**A**,**B**) display mixture plots that highlight the desired compromise region, located within the ternary mixing zone between the molecules. This zone represents the optimal conditions for achieving maximum AGI activity. Panel (**C**) further illustrates this relationship through a 2D mixture plot, pinpointing the specific proportions of the compounds required to reach the desired AGI IC_50_ value of 62.49 µg/mL.

**Figure 8 cimb-47-00512-f008:**
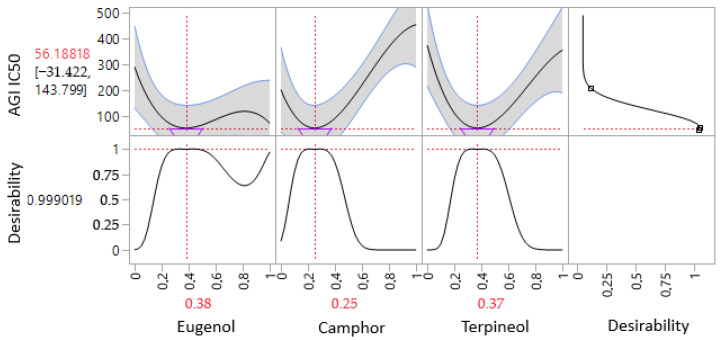
Desirability profile illustrating the precise proportions, leading to the optimum value for AGI IC_50_ of 56.18 µg/mL, was achieved with a mixture consisting of 38% eugenol, 25% camphor, and 37% terpineol. Gray area represents the confidence interval.

**Figure 9 cimb-47-00512-f009:**
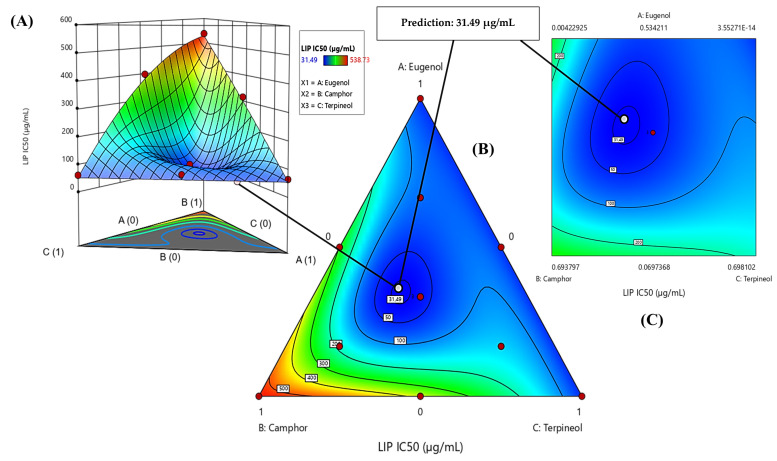
The optimal LIP IC_50_ value was determined through an in-depth analysis of 2D and 3D mixture plots, focusing on the identified compromise zone. Panels (**A**,**B**) display mixture plots that highlight the desired compromise region, located within the ternary mixing zone between the molecules. This zone represents the optimal conditions for achieving maximum LIP activity. Panel (**C**) further illustrates this relationship through a 2D mixture plot, pinpointing the specific proportions of the compounds required to reach the desired LIP IC_50_ value of 31.49 µg/mL.

**Figure 10 cimb-47-00512-f010:**
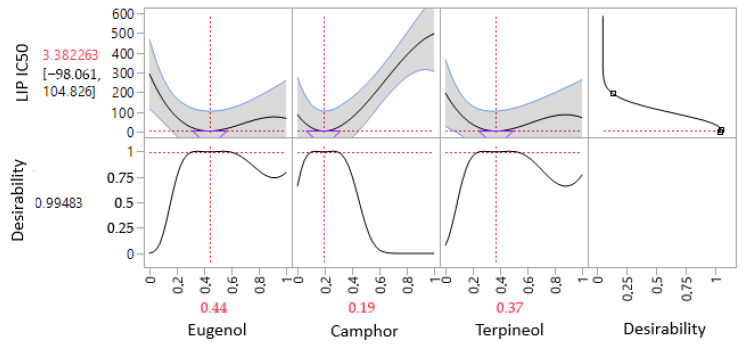
Desirability profile illustrating the precise proportions, leading to the optimum value for LIP IC_50_ of 3.38 µg/mL, was achieved with a mixture consisting of 44% eugenol, 19% camphor, and 37% terpineol. Gray area represents the confidence interval.

**Figure 11 cimb-47-00512-f011:**
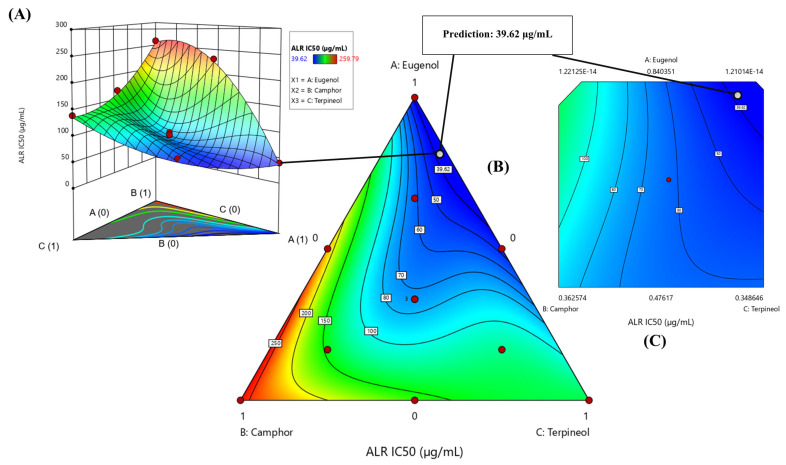
The optimal ALR IC_50_ value was determined through an in-depth analysis of 2D and 3D mixture plots, focusing on the identified compromise zone. Panels (**A**,**B**) display mixture plots that highlight the desired compromise region, located within the ternary mixing zone between the molecules. This zone represents the optimal conditions for achieving maximum ALR activity. Panel (**C**) further illustrates this relationship through a 2D mixture plot, pinpointing the specific proportions of the compounds required to reach the desired ALR IC_50_ value of 39.62 µg/mL.

**Figure 12 cimb-47-00512-f012:**
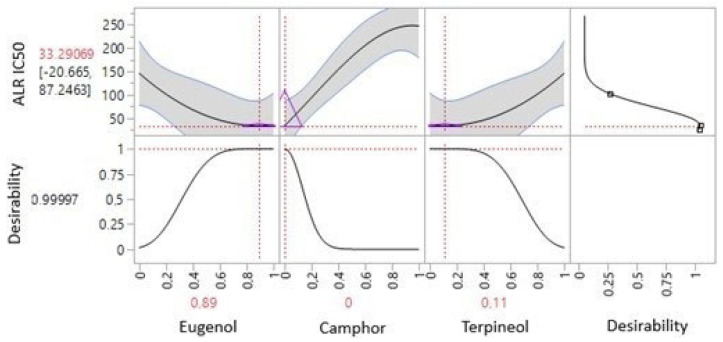
Desirability profile illustrating the precise proportions, leading to the optimum value for ALR IC_50_ of 33.29 µg/mL, was achieved with a mixture consisting of 89% eugenol, 0% camphor, and 11% terpineol. Gray area represents the confidence interval.

**Figure 13 cimb-47-00512-f013:**
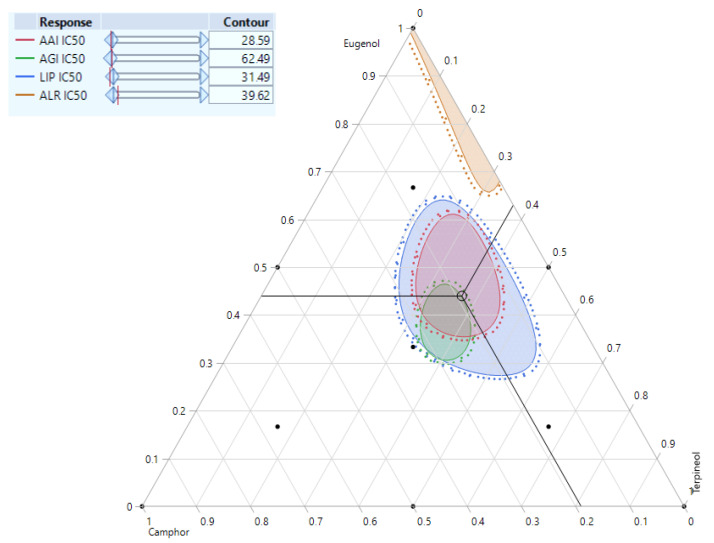
A 2D mixture contour plot of the optimal combination zone between the compounds, resulting in the best value of AAI, AGI, LIP, and ALR half inhibitory concentrations.

**Figure 14 cimb-47-00512-f014:**
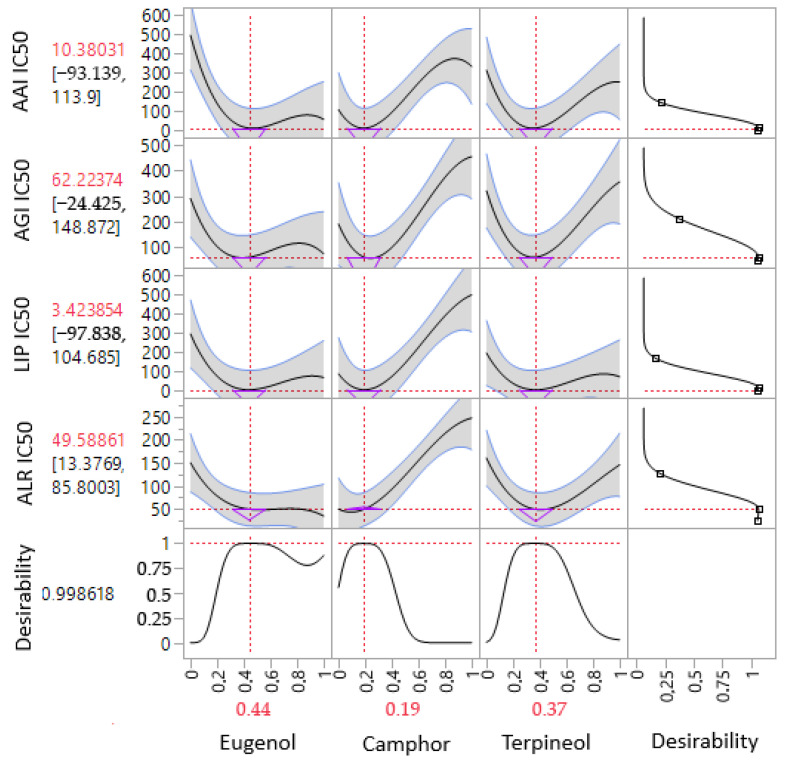
The desirability profiles for the simultaneous optimization of all responses identified an optimal mixture consisting of 44% eugenol, 0.19% camphor, and 37% terpineol. The best combination for the four studied responses resulted in the following optimal values: AAI IC_50_ at 10.38 µg/mL, AGI IC_50_ at 62.22 µg/mL, LIP IC_50_ at 3.42 µg/mL, and ALR IC_50_ at 49.58 µg/mL. Gray area represents the confidence interval.

**Figure 15 cimb-47-00512-f015:**
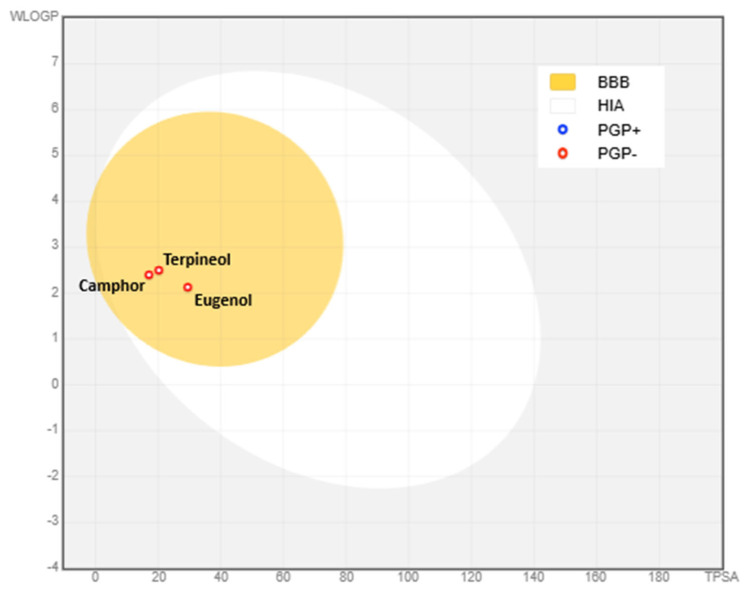
BOILED-EGG model of blood–brain barrier (BBB), and human intestinal absorption (HIA) of bioactive molecules.

**Table 1 cimb-47-00512-t001:** Identification of solvent system factors.

Components	Coded Variables	Level −	Level +
Eugenol	M1	0	1
Camphor	M2	0	1
Terpineol	M3	0	1
Sum of proportions		1	

**Table 2 cimb-47-00512-t002:** Antidiabetic mixture of the three compounds against four essential diabetes-linked enzymes.

No. ^a^	Eugenol	Camphor	Terpineol	Enzymatic Inhibition—IC_50_ (µg/mL) ^b^			
				AAI_IC50_	AGI_IC50_	LIP_IC50_	ALR_IC50_
1	1	0	0	55.63 ± 1.95	84.93 ± 6.14	63.69 ± 2.11	39.62 ± 1.78
2	0	1	0	355.82 ± 14.30	484.95 ± 17.65	538.73 ± 21.57	259.79 ± 8.65
3	0	0	1	267.97 ± 6.75	340.42 ± 23.19	60.42 ± 1.98	138.63 ± 2.09
4	0.50	0.50	0	420.49 ± 18.80	459.53 ± 26.98	315.85 ± 5.81	230.59 ± 5.60
5	0.50	0	0.50	131.85 ± 13.04	198.69 ± 8.48	71.93 ± 3.40	52.42 ± 3.21
6	0	0.50	0.50	569.09 ± 30.98	310.76 ± 13.63	398.12 ± 18.04	167.83 ± 2.58
7	0.333	0.333	0.333	92.85 ± 8.45	93.67 ± 3.47	62.55 ± 3.13	87.52 ± 1.29
8	0.333	0.333	0.333	80.69 ± 5.37	88.57 ± 9.85	71.42 ± 4.59	82.94 ± 4.08
9	0.333	0.333	0.333	117.50 ± 9.84	95.84 ± 5.09	63.74 ± 3.96	80.47 ± 6.86
10	0.667	0.167	0.167	28.59 ± 0.42	62.49 ± 2.18	31.49 ± 1.30	42.42 ± 2.09
11	0.167	0.667	0.167	183.49 ± 6.19	155.85 ± 3.13	131.78 ± 2.34	125.53 ± 7.65
12	0.167	0.167	0.667	84.59 ± 2.58	191.42 ± 1.34	82.53 ± 5.22	98.04 ± 2.13
Acarbose	-	-	-	39.63 ± 2.41	59.22 ± 1.94	-	-
Orlistat	-	-	-	-	-	76.89 ± 5.52	-
Quercetin	-	-	-	-	-	-	38.63 ± 1.63

^a^ Experiments were performed after randomization. ^b^ The tests were conducted in three independent replicates and established as means ± SD.

**Table 3 cimb-47-00512-t003:** Variance analysis for the three fitted models.

**AAI_IC50_**	**Model**	**DF**	**SS**	**MS**	**F**	** *p* ** **-value**
	R	6	283,358.31	47,226.4	7.5305	0.0213 *
	r	5	31,356.65	6271.3		
	PE	2	703.488	351.7		
	Total	11	314,714.96			
	R^2^	0.91				
	R^2^ adj	0.78				
**AGI_IC50_**	**Model**	**DF**	**SS**	**MS**	**F**	** *p* ** **-value**
	R	6	224,988.16	37,498.0	8.5343	0.0163 *
	r	5	21,968.97	4393.8		
	PE	2	27.857	13.93		
	Total	11	246,957.13			
	R^2^	0.96				
	R^2^ adj	0.85				
**LIP_IC50_**	**Model**	**DF**	**SS**	**MS**	**F**	** *p* ** **-value**
	R	6	271,228.68	45,204.8	7.5332	0.0213 *
	r	5	30,003.74	6000.7		
	PE	2	46.358	23.18		
	Total	11	301,232.41			
	R^2^	0.96				
	R^2^ adj	0.85				
**ALR_IC50_**	**Model**	**DF**	**SS**	**MS**	**F**	** *p* ** **-value**
	R	6	52,022.268	8670.38	11.2986	0.0088 *
	r	5	3836.934	767.39		
	PE	2	25.5933	12.80		
	Total	11	55,859.203			
	R^2^	0.97				
	R^2^ adj	0.89				

DF: degree of freedom; SS: sum of squares; MS: mean square; R: regression; r: residual; PE: pure error; R^2^: coefficient of determination; adj: adjusted; * Statistically significant at *p* < 0.05.

**Table 4 cimb-47-00512-t004:** Coefficients of the two presumed models and their level of significance (*p*-value).

Term	Coefficients	AAI _IC50_		AGI_IC50_		LIP_IC50_		ALR_IC50_	
		Estimation	*p*-Value	Estimation	*p*-Value	Estimation	*p*-Value	Estimation	*p*-Value
Eugenol (Mixture)	δ_1_	56.3055	0.4947	75.31413	0.2924	67.58667	0.4077	34.965319	0.2481
Camphor (Mixture)	δ_2_	331.7018	** 0.0074 * **	454.2423	** 0.0009 * **	497.1394	** 0.0012 * **	247.29442	** 0.0002 * **
Terpineol (Mixture)	δ_3_	251.1055	** 0.0219 * **	356.2650	** 0.0026 * **	71.05483	0.3858	145.84896	** 0.0028 * **
Eugenol × Camphor	δ_12_	812.1742	0.0888	617.7124	0.1135	−16.8283	0.9661	289.23923	0.0846
Eugenol × Terpineol	δ_13_	−152.178	0.7090	−43.4816	0.8980	68.56309	0.8627	−141.6914	0.3411
Camphor × Terpineol	δ_23_	946.8141	0.0573	−437.425	0.2328	332.2680	0.4182	−136.0733	0.3588
Eugenol × Camphor × Terpineol	δ_123_	−9041.406	** 0.0076 * **	−6541.029	** 0.0136 * **	−5825.262	** 0.0361 * **	−1897.183	** 0.0489 * **

* Bold red values indicate a statistical significance at *p* < 0.05.

**Table 5 cimb-47-00512-t005:** Expected and observed responses for the test point that the best-fit mixes were able to achieve.

Enzymes.		IC_50_ (µg/mL)	t-Statistic	*p*-Value	Proportions of Each Compound (%)		
					Eugenol	Camphor	Terpineol
α-amylase (AAI)	Predi. ^a^	9.60 ± 0.00	−2.5207	0.0653	47%	18%	35%
	Exp. ^b^	11.23 ± 1.12					
α-glucosidase (AGI)	Predi.	56.18 ± 0.00	2.7429	0.0517	38%	25%	37%
	Exp.	51.73 ± 2.81					
Lipase (LIP)	Predi.	3.38 ± 0.00	−2.3316	0.0801	44%	19%	37%
	Exp.	3.73 ± 0.26					
Aldose reductase (ALR)	Predi.	33.29 ± 0.00	−1.8759	0.1339	89%	0	11%
	Exp.	36.94 ± 3.37					

^a^ The experimental value is represented as the average of three replicates. ^b^ The expected value includes the response’s standard deviation (± SD), as determined by the model.

**Table 6 cimb-47-00512-t006:** Physico-chemical, drug-likeness, and the bioavailability of thymol, terpinene, and limonene.

Characteristics	Eugenol	Camphor	Terpineol
Structure	** 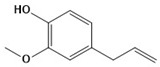 **	** 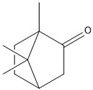 **	** 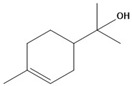 **
Molecular formula	C_10_H_12_O_2_	C_10_H_16_O	C_10_H_18_O
**Physico-chemical properties and drug-likeness**
MW (g/mol)	164.20	152.23	154.25
TPSA (Å^2^)	29.46	17.07	20.23
Num. H-Bond acceptors	2	1	1
Num. H-Bond donors	1	0	1
Rotatable bonds	3	0	1
LogP	2.13	2.40	2.50
Lipinski *	Yes, no violation	Yes, no violation	Yes, no violation
Egan **	Yes	Yes	Yes
Veber ***	Yes	Yes	Yes
**Bioavailability score**	0.55	0.55	0.55
Bioavailability radars	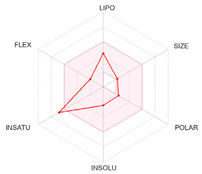	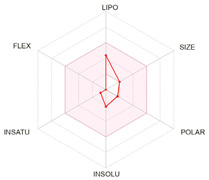	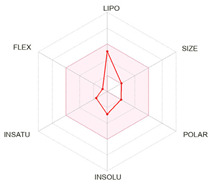

MW: Molecular weight; TPSA: Topological polar surface area; LogP: Octanol–Water Partition Coefficient; * Lipinski’s rule of five: MW ≤ 500; HBD ≤ 5; HBA ≤ 10; LogP ≤ 5. ** Egan rule: 0 ≤ LogP ≤ 5.88; TPSA ≤ 131 Å^2^. *** Veber rule: rotatable bonds ≤ 10; TPSA ≤ 140 Å^2^.

**Table 7 cimb-47-00512-t007:** In silico toxicological assessment of camphor and menthol using ProTox-III prediction tool.

Compound	Predicted LD_50_ (mg/kg)	Class	Nephrotoxicity	Hepatotoxicity	Carcinogenicity	Immunotoxicity	Mutagenicity	Cytotoxicity
Eugenol	1930	IV	Ina. (0.63)	Ina. (0.67)	Ina. (0.73)	Ina. (0.83)	Ina. (0.97)	Ina. (0.90)
Camphor	775	IV	Ina. (0.89)	Ina. (0.72)	Ina. (0.68)	Ina. (0.96)	Ina. (0.94)	Ina. (0.61)
Terpineol	2830	V	Ina. (0.89)	Ina. (0.72)	Ina. (0.76)	Ina. (0.99)	Ina. (0.90)	Ina. (0.64)

Ina.: Inactive; GHS: Globally Harmonized System; GHS hazard classes: IV: encompasses substances with oral toxicity. (LD50) ranging from 300 to 2000 mg/kg; V: identifies compounds with LD50 values between 2000 and 5000 mg/kg.

## Data Availability

The raw data supporting the conclusions of this article will be made available by the authors on request.

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
