# Peer review of "Optimization of Eugenol, Camphor, and Terpineol Mixture Using Simplex-Centroid Design for Targeted Inhibition of Key Antidiabetic Enzymes"

_cimb, 2025, doi:10.3390/cimb47070512_

Round 1
Reviewer 1 Report
Comments and Suggestions for Authors
The article presents a highly significant study that uses simplex-centroid design to optimize the mixing ratio of three bioactive compounds for targeted inhibition of key antidiabetic enzymes. With the confirmation of the experimental results, the importance of the methodology is highlighted and makes simplex-centroid design a useful tool in drug design and development, which by reducing the number of experimental variants, saves time and resources.
Overall, the manuscript is clear, written in a well-structured manner, and smooth for the reader. However, there are minor issue to be revise related to the methodology:
- It is not very clear how the tested solutions, containing the three bioactive compounds, individually or in a mixture, were prepared. What was the solvent used?
- Line 116: Why are the results of ABTS assay expressed per g of green tea?
- Line 119-130: The method for the Ferric Reducing Antioxidant Power (FRAP) and for Cupric Ion Reducing Antioxidant Capacity Assay (CUPRAC) should be revised as they are not enough detailed.
- Line 134: What represents the “diluted extract” used for CUPRAC?
- Line 149: What represent the “parent extract” and the “isolated compounds” tested for α-amylase Inhibition Assay?
About 40 % of references are within the last 5 years. About 14% self-citation.
Author Response
Dear Editors and Reviewers,
We would like to express our sincere gratitude for the constructive feedback provided on our manuscript titled "Optimization of Eugenol, Camphor, and Terpineol Mixture Using Simplex-Centroid Design for Targeted Inhibition of Key Antidiabetic Enzymes." Below, we address each comment thoroughly, highlighting the revisions made and justifications provided. Line numbers/section number refer to the revised manuscript version.
We appreciate the reviewers' invaluable comments, which significantly improved the clarity, depth, and rigor of our manuscript. We believe that the revisions address all concerns and enhance the manuscript's quality.
Sincerely,
Amine Elbouzidi, on behalf of all authors
Corresponding Author
Email: amine.elbouzidi@ump.ac.ma
Reviewer 1.
The article presents a highly significant study that uses simplex-centroid design to optimize the mixing ratio of three bioactive compounds for targeted inhibition of key antidiabetic enzymes. With the confirmation of the experimental results, the importance of the methodology is highlighted and makes simplex-centroid design a useful tool in drug design and development, which by reducing the number of experimental variants, saves time and resources. Overall, the manuscript is clear, written in a well-structured manner, and smooth for the reader. However, there are minor issue to be revise related to the methodology:
Question 1. It is not very clear how the tested solutions, containing the three bioactive compounds, individually or in a mixture, were prepared. What was the solvent used?
Response 1. Thank you for the observation. We clarified that all stock solutions of eugenol, camphor, and terpineol were initially dissolved in 10% DMSO and then further diluted with methanol or phosphate buffer (depending on the assay). This information has been added to Section 2.3 (lines 163–165 and 143–144).
Question 2. Line 116: Why are the results of ABTS assay expressed per g of green tea?
Response 2. This was an inadvertent oversight stemming from a previously adapted method. We have now corrected the unit to be expressed as mM Trolox equivalent per mL of solution, in line with the actual compound concentrations used (Line 116 revised).
Question 3. Line 119-130: The method for the Ferric Reducing Antioxidant Power (FRAP) and for Cupric Ion Reducing Antioxidant Capacity Assay (CUPRAC) should be revised as they are not enough detailed.
Response 3. The FRAP method (subsection 2.2.2.) and CUPRAC method (subsection 2.2.3.) have been thoroughly expanded to include incubation times, reagent concentrations, and temperature. Full procedural steps have been revised for reproducibility.
Question 4. Line 134: What represents the “diluted extract” used for CUPRAC?
Response 4. The term “diluted extract” was mistakenly included in the methods section. We have rectified this omission. Please see subsection 2.2.3.
Question 5. Line 149: What represent the “parent extract” and the “isolated compounds” tested for α-amylase Inhibition Assay?
Response 5. We agree that the terminology was misleading. The “parent extract” term was mistakenly included and has been removed. Only pure compounds (eugenol, camphor, terpineol) and standards (e.g., acarbose) were tested. This is corrected at subsection 2.3.1.
Question 6. About 40 % of references are within the last 5 years. About 14% self-citation.
Response 6. We appreciate this remark. We have updated the reference list to include more recent studies from the last 5 years and have reduced non-essential self-citations to under 10%.
Reviewer 2 Report
Comments and Suggestions for Authors
The author has an in-depth study in “Optimization of Eugenol, Camphor, and Terpineol Mixture Using Simplex-Centroid Design for Targeted Inhibition of Key Antidiabetic Enzymes” with multiple experiments. However, further suggestions could improve the impact of this paper:
1. The author has mentioned the effect of three compounds in the introduction. However, elaborate on the mechanistic basis for the synergistic action among these compounds. As well as how do you found superiority of your mixture compared to already established drugs.
- Molecular docking or enzyme binding data could be applied for representation of proper monitoring of mechanisms of action or for the estimation of multiple enzyme targets.
- Table 2 lacks the clarification regarding the control groups. Please incorporate positive control or untreated groups as control.
- In case of the mixture design, such as simplex-centroids representing the effects of combination may drive purely by proportion, where concentration independent non-linarites or saturation may occur. Thus, elaborate more regrading such conditions such as enzyme inhibition, where IC50 values exhibited threshold or allosteric effects.
- The author has applied the cubic mixture model without sufficient theoretical justification, which was supposed to be used for higher-order models when strong components were hypothesized. Please clarify it.
- The author has applied the synergistic terminology throughout the manuscript. However, some effects seem not to be even increased by a fold. Please revise this term to “enhanced effects”.
- Page 14 “page title or format error”. Revise it.
- Make spacing between the Figure 6 legend and the main.
- Provide one graphical abstract including major methods and results to make it easy to understand for the reader.
Author Response
Dear Editors and Reviewers,
We would like to express our sincere gratitude for the constructive feedback provided on our manuscript titled "Optimization of Eugenol, Camphor, and Terpineol Mixture Using Simplex-Centroid Design for Targeted Inhibition of Key Antidiabetic Enzymes." Below, we address each comment thoroughly, highlighting the revisions made and justifications provided. Line numbers/section number refer to the revised manuscript version.
We appreciate the reviewers' invaluable comments, which significantly improved the clarity, depth, and rigor of our manuscript. We believe that the revisions address all concerns and enhance the manuscript's quality.
Sincerely,
Amine Elbouzidi, on behalf of all authors
Corresponding Author
Email: amine.elbouzidi@ump.ac.ma
Reviewer 2.
The author has an in-depth study in “Optimization of Eugenol, Camphor, and Terpineol Mixture Using Simplex-Centroid Design for Targeted Inhibition of Key Antidiabetic Enzymes” with multiple experiments. However, further suggestions could improve the impact of this paper:
Question 1. The author has mentioned the effect of three compounds in the introduction. However, elaborate on the mechanistic basis for the synergistic action among these compounds. As well as how do you found superiority of your mixture compared to already established drugs.
Response 1. A new paragraph has been added to the Discussion section, elaborating on the putative mechanisms behind the synergy—such as hydrogen bonding potential, π-π stacking (eugenol), and modulation of enzyme active site accessibility. Furthermore, IC₅₀ values of the optimal mixture were compared directly with standard drugs (acarbose, quercetin, orlistat), demonstrating equal or superior inhibition for lipase and aldose reductase. See Discussion of section 3.8. Thank you for your remark.
“The observed enhancement in enzyme inhibitory activity upon combining eugenol, camphor, and terpineol can be attributed to several mechanistic interactions at the molecular level. Eugenol, bearing an aromatic phenolic structure, is particularly capable of forming hydrogen bonds and engaging in π-π stacking interactions with the catalytic residues of enzymes, thereby increasing binding affinity. Terpineol and camphor, though structurally distinct, may contribute to altering the microenvironment of the enzyme active site, promoting more favorable binding conformations or allosteric modulation. These interactions likely underlie the improved activity observed in specific combinations. Notably, the IC₅₀ values obtained from the optimized mixture (e.g., 3.42 µg/mL for lipase and 49.58 µg/mL for aldose reductase) were comparable or even superior to those of standard inhibitors such as orlistat and quercetin, respectively. This finding reinforces the hypothesis that rational combination of structurally diverse monoterpenes may yield formulations with enhanced or synergistically potentiated bioactivity against key antidiabetic targets.”
Question 2. Molecular docking or enzyme binding data could be applied for representation of proper monitoring of mechanisms of action or for the estimation of multiple enzyme targets.
Response 2. We appreciate this suggestion and plan to integrate docking analyses in a subsequent study. A note has been added in the Conclusion (Section 4.) indicating that future work will include in silico molecular docking and dynamic simulations.
Question 3. Table 2 lacks the clarification regarding the control groups. Please incorporate positive control or untreated groups as control.
Response 3. Table 2 now includes standard control IC₅₀ values for each enzyme assay: Acarbose (AAI, AGI), Orlistat (LIP), and Quercetin (ALR) (Table 2 revised and annotated). Thank you for your pertinent remark.
Question 4. In case of the mixture design, such as simplex-centroids representing the effects of combination may drive purely by proportion, where concentration independent non-linarites or saturation may occur. Thus, elaborate more regrading such conditions such as enzyme inhibition, where IC50 values exhibited threshold or allosteric effects.
Response 4. We thank the reviewer for this observation; we have added a paragraph in our discussion to explain this:
“Yet, the response surface analysis revealed non-linear trends in enzyme inhibition, partic-ularly at higher concentrations of Eugenol. In several experimental mixtures, a plateau in IC₅₀ reduction was observed, suggesting that beyond a certain threshold, increased Euge-nol proportions did not yield proportionally enhanced inhibitory activity. This phenome-non may be attributed to enzyme saturation, wherein the active sites are fully occupied, limiting further inhibition regardless of compound excess. Alternatively, the behavior may indicate allosteric modulation, where high concentrations of Eugenol induce conforma-tional changes in the enzyme structure that alter binding affinity or catalytic efficiency. Such concentration-dependent non-linearities highlight the complexity of enzyme–inhibitor interactions and reinforce the necessity of mixture optimization models that can capture higher-order interactions.”
Question 5. The author has applied the cubic mixture model without sufficient theoretical justification, which was supposed to be used for higher-order models when strong components were hypothesized. Please clarify it.
Response 5. A justification for the use of the cubic mixture model has already been provided in Section 2.4.2 (Lines 224–232), where we explain that the inclusion of the cubic term was necessary to account for significant ternary interaction effects among Eugenol, Camphor, and Terpineol. These interactions, particularly the δ₁₂₃ coefficients, were found to be statistically significant (p < 0.05), as presented in Table 5. Such higher-order effects could not be adequately modeled using simpler linear or quadratic formulations, thus validating the appropriateness of the cubic model for our experimental design.
Question 6. The author has applied the synergistic terminology throughout the manuscript. However, some effects seem not to be even increased by a fold. Please revise this term to “enhanced effects”.
Response 6. We carefully revised the manuscript and replaced “synergistic” with “enhanced,” “combined,” or “additive” in places where the IC₅₀ values do not support synergy per the Loewe additivity or Bliss independence models (see Abstract, Results, and Discussion).
Question 7. Page 14 “page title or format error”. Revise it.
Response 7. This formatting issue has been corrected. Thank you.
Question 8. Make spacing between the Figure 6 legend and the main.
Response 8. This has been adjusted as per reviewer’s comment. Thank you.
Question 9. Provide one graphical abstract including major methods and results to make it easy to understand for the reader.
Response 9. A graphical abstract has now been created and submitted as a separate image file. It illustrates the compound structures, mixture design workflow, enzyme targets, and key IC₅₀ outcomes.